# Unveiling gene perturbation effects through gene regulatory networks inference from single-cell transcriptomic data

Clelia Corridori[1], Merrit Romeike[2,3,4,5,6], Giorgio Nicoletti[7,8], Christa Buecker[2,3], Samir Suweis[9]*, Sandro Azaele[9]*, Graziano Martello[1]*

**1** Department of Biology, University of Padua, Padova, Italy, **2** Max Perutz Labs, Vienna Biocenter Campus (VBC), Vienna, Austria, **3** Center for Molecular Biology, Department of Microbiology, Immunobiology and Genetics, University of Vienna, Vienna, Austria, **4** Vienna Biocenter PhD Program, Doctoral School of the University of Vienna and the Medical University of Vienna, Vienna, Austria, **5** Max Planck Institute for Molecular Genetics, Berlin, Germany, **6** Berlin Institute of Health, BIH Center for Regenerative Therapies, Charité-Universitätsmedizin, Berlin, Germany, **7** Quantitative Life Sciences Section, The Abdus Salam International Center for Theoretical Physics (ICTP), Trieste, Italy, **8** ECHO Laboratory, Ecole Polytechnique Fédérale de Lausanne, Lausanne, Switzerland, **9** Department of Physics and Astronomy "G. Galilei", University of Padua, Padova, Italy

* graziano.martello@unipd.it (GM); samir.suweis@unipd.it (SS); sandro.azaele@unipd.it (SA)

## Abstract

Physiological and pathological processes are governed by networks of genes called gene regulatory networks (GRNs). By reconstructing GRNs, we can accurately model how cells behave in their natural state and predict how genetic changes will affect them. Transcriptomic data of single cells are now available for a wide range of cellular processes in multiple species. Thus, a method building predictive GRNs from single-cell RNA sequencing (scRNA-seq) data, without any additional prior knowledge, could have a great impact on our understanding of biological processes and the genes playing a key role in them. To this aim, we developed IGNITE (Inference of Gene Networks using Inverse kinetic Theory and Experiments), an unsupervised machine learning framework designed to infer directed, weighted, and signed GRNs directly from unperturbed single-cell RNA sequencing data. IGNITE uses the GRNs to generate gene expression data upon single and multiple genetic perturbations. IGNITE is based on the inverse problem for a kinetic Ising model, a model from statistical physics that has been successfully applied to biological networks. We tested IGNITE on two complementary systems of pluripotent stem cells (PSCs): murine PSCs transitioning from the naïve to formative states, and human PSCs differentiating toward definitive endoderm. These datasets differ in species, developmental trajectory, and single-cell technology (10X vs. Fluidigm C1), providing a stringent test of generalizability. Using only unperturbed scRNA-seq data, IGNITE simulated single and multiple gene knockouts (KOs) and produced predictions consistent with independent experimental observations. In mouse PSCs, IGNITE generated wild-type data highly correlated with experiments and accurately predicted the effects of Rbpj,

**Data availability statement:** The code implementing the IGNITE framework and all analysis scripts used in this study are publicly available at https://github.com/CleliaCorridori/IGNITE. The repository also includes the preprocessed datasets used in the analyses for both the mouse and human PSC datasets (log-normalized matrices with PST and MB), enabling full reproduction of the results presented in this work. Raw scRNA-seq data are available at ENA with ID: PRJEB74673.

**Funding:** C.C. was supported by a Microsoft Research PhD studentship (MRL 2020-035). C.B. was supported by Austrian Science Fund (FWF, Grant DOI: 10.55776/P34123). S.A. acknowledges financial support under the National Recovery and Resilience Plan (NRRP), Mission 4, Component 2, Investment 1.1, funded by the European Union – Next Generation EU – "Emergent Dynamical Patterns of Disordered Systems with Applications to Natural Communities" – CUP 2022WPHMXK. S.S. acknowledges DFA UNIPD for funding PARD2024 "Response theory for brain network discovery and control". G.M.'s Laboratory is supported by grants from the Giovanni Armenise–Harvard Foundation (Dissecting the Human Pluripotency Network), the Telethon Foundation (GJC21157), an ERC Starting Grant (MetEpiStem), the Progetti di Rilevante Interesse Nazionale PRIN 2022 (Dissecting genetic, epigenetic and metabolic alterations caused by reprogramming of somatic cells to pluripotency), the Microsoft Research Ltd Grant (The Pluripotency Program in Human Embryonic Stem Cells), the HUMANIZE Project la CAIXA Foundation (Generation of humanized organs from human iPS cells) and by the European Union – Next Generation EU, Mission 4 Component 1, CUP C93C22002780006, Spoke n.3 (AAV-delivered MTF1 to suppress suppressors of polyQ toxicity). The funders had no role in study design, data collection and analysis, decision to publish, or preparation of the manuscript.

**Competing interests:** The authors have declared that no competing interests exist.

Etv5, and triple KOs, while in human PSCs it correctly predicted differentiation-promoting and differentiation-blocking perturbations, in agreement with published studies. These results demonstrate that IGNITE robustly captures gene interaction logic across species and technologies, enabling robust in silico perturbation analyses directly from scRNA-seq data.

## Author summary

Physiological and pathological processes rely on complex interactions between genes, organized into gene regulatory networks that control how cells behave and respond to change. Understanding these networks is essential to predict how genetic perturbations influence cell states, but experimental testing of gene function is often time-consuming, costly, or technically limited. Computational approaches can help address this challenge, although many existing methods require additional experimental data or prior biological knowledge. Here, we introduce IGNITE, an unsupervised machine learning framework that infers gene regulatory networks directly from unperturbed single-cell gene expression data. Without relying on prior information, the method reconstructs effective gene interactions and uses them to simulate how cells respond to genetic perturbations, such as gene knockouts. We applied this approach to two distinct biological systems: mouse naive pluripotent stem cells transitioning to the formative state, and human pluripotent stem cells differentiating toward endoderm. Despite differences in species, developmental trajectories, and experimental technologies, the model accurately reproduced observed cell states and predicted the effects of genetic perturbations consistent with independent experiments. Overall, this work introduces a general and scalable machine learning framework for building predictive models of gene regulation directly from single-cell data.

## 1 Introduction

The correct functioning of cellular processes requires regulated interactions of several components. By studying each individual element separately, we cannot grasp how the organisation of the individual components involved in a specific process can lead to complex behaviors. However, studying the whole system is still a challenging task. Single-cell RNA sequencing (scRNA-seq) data [1,2] have the potential to unravel different biological processes [3], and modelling them as complex systems using a statistical physics description can provide deeper insights into their underlying mechanisms [4–6].

Among all the elementary components of the cell, genes are the key necessary elements in regulating biological processes by interacting and organising themselves into Gene Regulatory Networks (GRN). To study the interactions between genes and how they shape the cell, it is possible to experimentally perturb the regulation of

individual genes, forcing their activation or deactivation. An experimental technique to perturb the GRN by deactivating specific genes is the knockout. This technique allows to measure the effects of individual or multiple gene deactivations on the regulation of other genes, and how this affects phenomenological cell dynamics. However, given the multitude of genes on which it is possible to focus, an experimental approach would be too time-intensive and costly. In silico models represent a viable alternative, as they can infer complex gene interactions and predict the outcome of genetic perturbation, ultimately revealing key regulatory mechanisms.

Emerging computational strategies aim to predict how perturbations affect cell phenotypes. To effectively apply these methods, it is essential to understand their capabilities and limitations. For instance, the reliance on data obtained from experimental perturbations for the training step limits the applicability of these models [7]. Furthermore, several approaches require the integration of multi-omics data as prior knowledge for GRN inference [8,9]. This implies having access to these data by performing different measurements on the same biological system. Alternatively, some methods can infer GRN interactions only from unperturbed single-cell RNA sequencing data [10]. However, they focus only on some specific aspects, such as GRN inference, or generation of wild-type (WT) data, or perturbation simulations.

Among the approaches for GRN inference, RE:IN [11–14] is capable of inferring interactions among numerous genes using a Boolean network modelling approach and simulating knockouts of one or multiple genes. However, RE:IN relied on both knockout experiments and bulk transcriptomic data, lacking single-cell resolution, and needed to include other types of data (ChIP sequencing data, named ChIP-seq, and literature-curated interactions) to constrain the model. Moreover, RE:IN should explore a number of possible networks that increase exponentially with the number of genes, making the inference task computationally demanding. Another method of inferring the GRNs is MaxEnt [15]. It employs the maximum entropy principle to infer genetic interaction networks from gene expression patterns. The GRN is obtained by maximising the entropy while satisfying the specific constraints given by the data [16,17]. However, MaxEnt can infer only undirected interactions, and it is not suitable for generating perturbed data. SCODE [18] is one of the gold standard models for inferring directed weighted networks starting from scRNA-seq data and exploiting pseudotime. It describes gene expression dynamics with a set of coupled linear ordinary differential equations. Moreover, it can generate new data, but it has not been validated for the simulation of gene perturbation. CellOracle [8] is a recent network-based machine learning framework. It uses scRNA-seq data from different clusters of cells and map active regulatory connections onto a GRN framework computed using scATAC-sec data (or ATAC-seq data). It produces GRNs for each cluster of the dataset and simulates the effects of perturbing a transcription factor with a known binding motif. CellOracle focuses on the role of GRN perturbation in modifying cell phenotypes, without describing how GRN can generate unperturbed phenotypes.

Therefore, all these approaches study only some specific features that resemble the real system, which we summarised in Table 1. However, it would be essential to develop interpretable models that require only one type of unperturbed data (e.g., single cell transcriptomics), simplifying algorithm usage and providing a description that includes both

**Table 1. Comparison of different computational methods for GRN inference, highlighting their input data type, requirement of prior information, capabilities of generating new data, implementation of multiple gene knockouts (KO) simulations, and ability to infer directed interactions. An asterisk (*) indicates that the feature is theoretically supported by the method but was not implemented and discussed in the referenced publication.**

| Methods | Input data resolution | Prior information | Generative model | Multiple KO | Directed | PST |
|---|---|---|---|---|---|---|
| RE:IN | Bulk RNA-seq | ChIP-seq data and literature-curated interactions | Yes | Yes | Yes | No |
| MaxEnt | Bulk or scRNA-seq | Only transcriptomics | No* | No | No | No |
| SCODE | scRNA-seq | Only transcriptomics | Yes | No* | Yes | Yes |
| CellOracle | scRNA-seq | scATACseq or ATAC-seq | No | No* | Yes | Yes |
| IGNITE | scRNA-seq | Only transcriptomics | Yes | Yes | Yes | Yes |

GRN inference, a generative rule explaining how GRN can produce cell phenotypes, with the capacity to simulate the effects of gene perturbations.

In response to these modelling challenges, we propose IGNITE (Inference of Gene Networks using Inverse kinetic Theory and Experiments). It is an unsupervised computational approach, based on concepts of statistical physics, that starts from unperturbed scRNA-seq data to infer GRN models capable of simulating unperturbed or perturbed cell phenotypes. In particular, it relies on the inverse problem for a kinetic Ising model [19], following the approach proposed by Zeng et al. in [20]. The Ising model was initially developed to describe ferromagnetism and, by using an inverse approach, has found applications in modelling several many-body systems, including biological networks and neural networks [15,21,22]. Here, we leverage its versatility to develop a computational model describing the GRN by exploiting all the features reported in Table 1.

We applied IGNITE to the study of the developmental progression of pluripotent stem cells (PSCs). Pluripotency is the potential to give rise to all cells found in the adult and is observed in the naïve epiblast of the preimplantation embryo. Upon embryo implantation, epiblast transition to a different pluripotency phase, termed formative phase, characterised by distinct gene expression profiles, epigenetic state and developmental features. During this transition, naïve pluripotency markers are down-regulated, epigenetic modifiers and formative markers are activated, and cells become competent in the formation of the germ layer and the specification of primordial germ cells (E6.5–7.5) [14]. Pluripotent cells of the early embryo can be captured in vitro as naïve Embryonic Stem cells (ESC), which retain pluripotency and the capacity to progress from the naïve to the formative state. Several genetic approaches have been used to identify genes crucial for the maintenance of the naïve state, or the transition to the formative state. However, a global understanding of how the transition is regulated is still missing.

Previous approaches focused mainly on characterising the state of naïve pluripotency and elucidating the functional interactions among the molecular components that govern and sustain it [23–26]. Moreover, efforts focused on modelling the interactions of these core genes and their dynamics under different cell culture conditions. For instance, Walker et al. [23] experimentally deduced the possible GRNs of mouse ESCs in the pluripotency state, while other approaches, relying on established GRNs, characterised the dynamics of a few interacting genes using differential equations [24–26]. However, while informative and precise, these approaches were limited to describing the interactions of a limited number of genes.

The Boolean network approach has been applied not only to bulk RNA-seq data (as done with RE:IN), but also to single-cell data [27–30]. These methods allowed to have single cell resolution data, but still required the integration of different types of data, (e.g., ChIP-Seq and knockout experiments) or should explore a number of possible networks that increase exponentially with the number of genes, making the inference task computationally demanding. Other computational strategies, including tree-based models, correlation and partial correlation analysis, information theory, and differential equations, have been used to model differentiation from unperturbed single-cell multi-omics data [10]. However, even in studies on differentiation, these methods focus primarily on inferring GRNs, as previously highlighted, without elucidating how these GRNs can generate differentiating cells or inspecting the role of perturbations in differentiation. Other methods used single-cell multi-omics data to infer the GRN and inspect the effect of possible perturbations without a theoretical description of the model that can generate the inferred GRNs [8,9].

Hence, it is essential to study this system to have a reliable model capable of generating all the observed cell phenotypes and being able to simulate knockout to investigate gene function during the transition. We applied IGNITE to two complementary systems of pluripotent stem cells. First, using a novel 10X scRNA-seq dataset of murine ESCs progressing from the naïve to the formative state, IGNITE inferred the GRN. From the GRN, IGNITE generated new synthetic data without depending on biological prior knowledge, such as different data types (e.g., ATACseq data) or literature-based information (e.g., established interactions). Remarkably, the generated wild-type data were statistically comparable with experimental measurements. IGNITE also simulated the knockout of three genes, Rbpj, Etv5, and Tcf7l1, both individually

and concurrently. These simulations were largely supported by experimental data. Of note, Tcf7l1 is a transcription factor regulated at the level of protein stability rather than transcriptionally; simulations of its inactivation by IGNITE and other methods were not accurate, revealing limitations of transcriptomics-only approaches.

Second, to further assess the generalizability of IGNITE, we applied it to an independent scRNA-seq dataset of human PSCs differentiating toward definitive endoderm from Chu et al. [31]. This dataset, previously analyzed for GRN inference by other groups [10,18], was generated using the Fluidigm C1 platform, which differs from the 10X technology employed in our mouse data and captures fewer cells. Beyond representing a distinct developmental trajectory, this dataset thus provides a stringent benchmark to test IGNITE across species, cell numbers, and experimental technologies. IGNITE inferred a biologically coherent GRN, reproduced wild-type differentiation dynamics, and correctly predicted differentiation-promoting and differentiation-blocking perturbations, in agreement with published studies.

In summary, IGNITE represents a model-based computational approach to generate new single-cell phenotypes as patterns of (de)activation of genes and to simulate perturbation experiments thanks to the inference of the GRN describing the biological system studied, outperforming current gold standard methods [8,18].

## 2 Results

### 2.1 IGNITE algorithm

The IGNITE algorithm consists of three main steps: preprocessing, modeling, and data generation (Fig 1A). (i) **Preprocessing**: scRNA-seq input data undergo quality control and logarithmic normalization, followed by pseudotime (PST) computation and application of the Mini-Bulk (MB) approach (see Methods section). The PST computation requires data dimensionality reduction and clustering steps. The PST algorithm Slingshot [32] orders individual cells along a pseudotime trajectory, assigning to each cell a pseudotime value that represents its position along that trajectory [33].

This implementation allows us to group transcriptionally similar cells before averaging. Dropout is a well-documented challenge in scRNA-seq data [34]. To mitigate this issue, we generated Mini-Bulks (MBs) by averaging neighboring cells along the trajectory. By construction, this procedure reduces the number of zero entries compared to single-cell profiles, thereby alleviating the impact of dropout. Applying PST and MB strategy avoids averaging across heterogeneous cells that, despite sharing the same sampling time, occupy different positions along the differentiation trajectory. The final step of the preprocessing is the binarization because IGNITE models gene expression as on/off spins states within a kinetic Ising model framework, to infer the unknown pairwise gene-gene interactions [15,20]. Therefore, the algorithm assigns state +1 to active genes (high gene expression) and −1 to inactive genes (low gene expression), obtaining Gene Activity (GA, see Methods for details). (ii) **Modelling**: The used system dynamical rule follows the Glauber dynamics with asynchronous spin updates [19]. IGNITE solves the inverse Ising problem by maximising the likelihood of the system [15,20]. In doing so, IGNITE generates multiple candidate GRNs, and, among them, selects the one with the lowest distance between input and generated data correlation matrices (Correlation Matrices Distance, CMD). (iii) **Data Generation**: The inferred GRN is then used to generate new data under desired conditions, either wild-type (WT) or gene perturbations (knockout, KO). Outputs include (1) the effective inferred GRN, (2) the generated GA under wild-type, and (3) the effects of specific gene perturbations on the gene activity (see Methods for details).

### 2.2 Deciphering pluripotent stem cell dynamics through scRNA-seq

We analyzed mouse (mPSC) and human pluripotent stem cells (hPSC) to characterize regulatory dynamics underlying cell-fate transitions. In this section, we focus on mPSCs, while later we extend the analysis to hPSCs to test IGNITE across species and experimental platforms. Our mPSC dataset captures the transition from the naïve to the formative state upon 2iL withdrawal [35]. This chemically defined culture condition ensures the naïve pluripotent state of stem cells [36,37]. Upon 2iL withdrawal, we profiled gene expression at multiple time points by scRNA-seq to monitor the dynamics

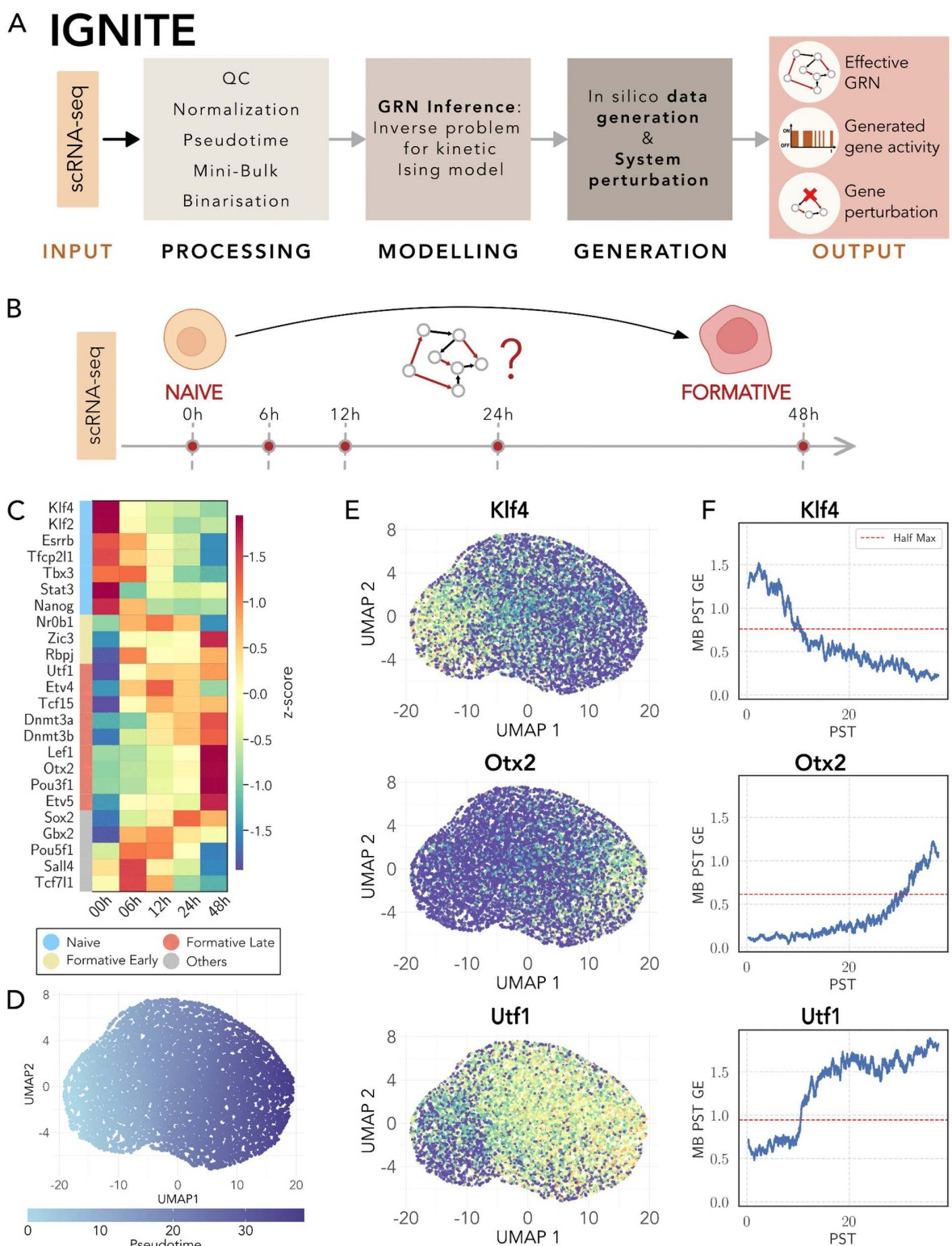

**Fig 1. IGNITE framework and mouse scRNA-seq dataset analysis. A.** Overview of the IGNITE workflow: from the scRNA-seq input, through processing, modelling, and data generation, to the final outputs: (i) effective inferred Gene regulatory Network (GRN), (ii) wild-type generated gene activity

data, and (iii) gene activity generated using inferred GRNs with gene perturbations. QC indicates Quality Control. **B.** Studied biological system: temporal dynamics of PSCs differentiation, encompassing the transition through distinct stages from naïve to formative. The considered transcriptomic dataset is 10X scRNA-seq starting in 2i+LIF (2iL). The points represent 5 time points at which specific cells were sampled and measured after the removal of 2iL. **C.** Average gene expression (GE) z-score values from the log-normalized scRNA-seq input dataset. For each gene, the average is computed for all cells that belong to the same time point. We considered 24 genes, divided into four different gene categories, naïve (light blue), formative early (yellow), formative late (red), and others (grey). We classified these genes following Carbognin et al. [14]. **D.** Two-dimensional UMAP visualisation of the log-normalized scRNA-seq dataset, with each point representing an individual cell. The gradient colour scale corresponds to the pseudotime value of each cell, indicating the time progression within the dataset. The pseudotime ranges from the value of 0 to the final value of 38.22. UMAP1 and UMAP2 indicate the two dimensions of the UMAP space. **E.** UMAP visualisation of three key genes (Klf4, Otx2, Utf1) in the log-normalized scRNA-seq dataset. Each point represents an individual cell, coloured by the normalized expression of the respective gene. **F.** Mini-Bulk gene expression of Klf4, Otx2, and Utf1 along pseudotime (PST). The red dashed line indicates half-maximum expression.

of gene activity during differentiation (Fig 1B). We focused on 24 genes known in the literature for their driving role in the decision-making process of differentiating PSCs [11,14,38,39], analyzing the first 48 hours of mPSC development. This time window allows to study the interactions that guide the transition between naïve and formative cells, analyzing a total of 9894 cells.

Gene expression profiles across pseudotime showed four temporal patterns (Fig 1C): naïve genes with higher activity in the naïve state that subsequently decreases, formative early genes activated shortly after 2iL withdrawal, formative late genes activated at later stages, and a mixed group without a monotonic trend, as previously reported [14].

To compute the PST, we first evaluated Principal Component Analysis (PCA) on a selected gene set (S1A and S1B Fig, see methods), and then projected the data using t-distributed Stochastic Neighbor Embedding (t-SNE) [40] and Uniform Manifold Approximation and Projection (UMAP) [41] (S1C Fig) and we clustered the data (S1D Fig), see Methods for details. To validate the biological significance of the obtained clusters, we investigated the distribution of gene expression in the genes within each cluster (refer to S1E Fig for three examples). In addition, we excluded cluster 1 from subsequent analysis due to its predominant composition of 2Cell-like cells (2CLCs), identified with confidence through a comprehensive analysis of differential gene expression [42]. We found as up-regulated markers genes such as Eif1a-like genes and Zscan4. This cluster was omitted from further investigation as it diverged from the main scope of our analysis, namely the transition from the naïve to the formative state. After computing the PST, we checked the PST value distribution for each time point (S1F Fig), observing agreement between sampling time and PST ordering. We verified that the expected temporal expression patterns of the genes were maintained even after PST ordering. In the low-dimensional UMAP space (Fig 1D), it is possible to observe the PST assigned to each cell. Cells with lower PST exhibit higher gene expression of naïve genes (e.g., Klf4, in Fig 1E) compared to formative genes. Conversely, formative genes displayed increased activity in cells with higher pseudotime values, as expected (e.g., Otx2 and Utf1 in Fig 1E). Gene expression dynamics, after PST and MB processing, is preserved. Indeed, its behavior over time (Figs 1F and S1G) is analogous to that described in Fig 1C. Finally, we calculated GA and still observed the expected transitions from the naïve to the formative states (S1H Fig).

### 2.3 Mouse GRN inference and WT data generation: IGNITE and SCODE

IGNITE uses only scRNA-seq data as input to infer a set of 250 GRNs (Fig 1A). It employs the correlation matrix of the generated data and its time derivative to infer the most accurate and effective GRN. It achieves this by comparing the correlation matrices between the input (processed scRNA-seq data with PST and MB) and generated datasets, quantifying their difference by means of their Euclidean distance. We defined this quantity as Correlation Matrices Distance, CMD (see Methods for details). The GRN with the lowest CMD (0.43) (Fig 2A–2C), which best reproduces the initial data correlations, is selected. Statistical validation by a two-sample t-test yielded a t-statistic $t = -43.49$ and a p-value $p < 2.2 \times 10^{-16}$, confirming a significant difference between the IGNITE model generated CMD and the null model CMD.

To test the capacity of IGNITE in generating data similar to the input ones, we computed the Spearman correlation between the average original values per gene and the average of the predicted ones. The resulting Spearman correlation

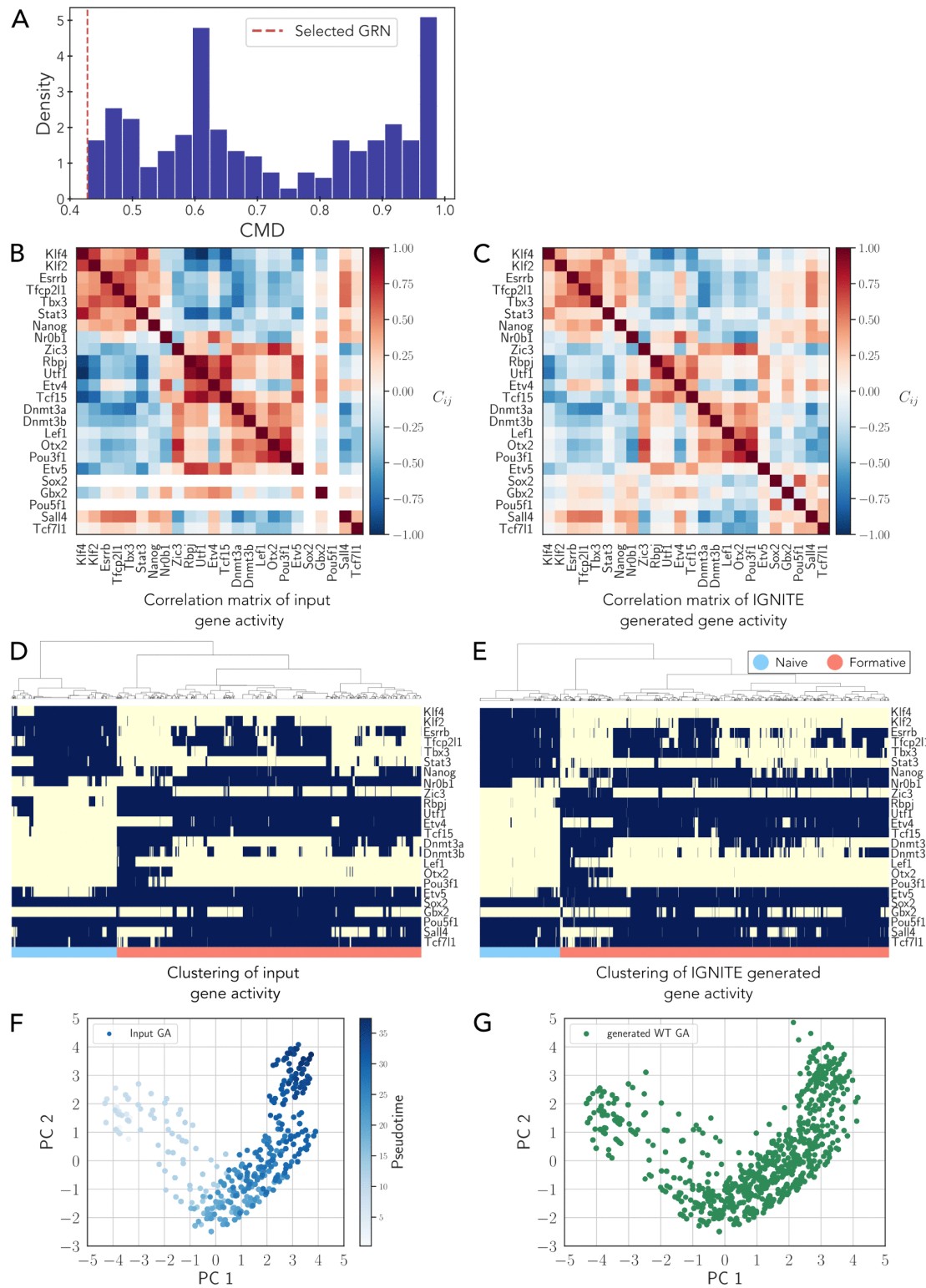

**Fig 2. Selection of the best-performing mouse GRN. A.** Density plot of the Correlation Matrices Distance (CMD) values for the 250 inferred GRNs. The red dashed line represents the selected GRN with the lowest CMD. **B.** Pearson correlation matrix for the gene activity (GA) of the input dataset

(scRNA-seq data with LogNorm, PST, and MB). **C.** Pearson Correlation matrix for IGNITE-generated gene activity dataset. **D.** Hierarchical clustering of gene activity of the input dataset. The clustering algorithm used is Ward's method. Each row represents a gene, while each column corresponds to an individual cell. The color indicates inactive (−1, yellow) or active (+1, blue) gene activity. The dataset has 9547 cells. **E.** Hierarchical clustering of gene activity of IGNITE-generated data. Methodology for visualisation as in Fig 2D. 9547 cells were simulated. **F.** Principal Component Analysis (PCA) scatter plot representing the gene activity, GA, for the input dataset (scRNA-seq data with LogNorm, PST, and MB). Each point corresponds to a single cell, and the colour intensity reflects the pseudotime value of the cell. PC1 and PC2 indicate the two dimensions of the PCA space. **G.** PCA scatter plot representing the IGNITE-generated wild-type gene activity data, WT GA, using the same dimensional reduction approach as in Fig 2F.

coefficient for the mean gene activity values is 0.99 (p-value <2.2 × 10⁻¹⁶, Spearman rank correlation test), overall indicating a high inter-gene correlation between input and generated data.

Furthermore, we evaluated the quality of IGNITE-generated data by clustering the original and generated binarised datasets. In the input dataset, naïve and formative clusters account for 26% and 74% of cells, respectively (Fig 2D). In the IGNITE-generated data, the corresponding fractions are 20% and 80% (Fig 2E). In the same PCA space, input and generated WT datasets occupy overlapping regions (Fig 2F and 2G).

Next, we compared the results obtained with IGNITE with those generated by SCODE. To infer the GRN with SCODE, we searched for the optimal parameters as suggested by the authors in [18] (S2A and S2B Fig, see Methods for details).

Looking at the clustering of the SCODE gene expression data, we observed that there are similarities between the input and the generated data (S2C and S2D Fig, respectively), with elevated expression of some naïve genes (e.g., Klf2/4) in naïve cells, and of formative genes Rbpj and Utf1 in formative cells. The comparison between input and SCODE-generated data reveals an overall agreement, as observed in the PCA representation (S2E Fig). Analysis of the correlation matrices yielded a CMD of 0.51 for SCODE, corresponding to a statistically significant deviation from the null model ($z$-score = −428.30). This value was higher than the CMD obtained with IGNITE (0.43), while both methods reproduced key features of the wild-type gene activity patterns.

## 2.4 Simulating gene knockouts in mPSCs with IGNITE

To assess the predictive capabilities of IGNITE, we simulated the knockout Rbpj, Etv5, and Tcf7l1 genes individually (single KOs) as well as the knockout of these genes simultaneously (triple KO) by removing their interactions from the effective GRN. Using these different KO GRNs, we generated the knockout datasets (see Methods for details).

To assess KO effects, we quantified changes in gene activity relative to wild-type simulations by measuring the difference between average gene activity in KO and WT conditions. Single KO simulations are characterised by increased activity for naïve genes and reduced for formatives, in agreement with previous reports [38,39]. For the triple knockout of Rbpj, Etv5, and Tcf7l1, both the simulations and experimental data displayed a strong increase in naïve genes and a decrease in formative genes.

To quantify these changes, we calculated the difference in gene activity/expression between KO and WT, scaled to the highest absolute value (scaled KO-WT difference, Fig 3A and 3B). We then compared the scaled KO–WT difference of generated data with the one obtained from experimental KO data retrieved from the literature [38,39]. The experimental scaled KO–WT difference measure is obtained by scaling the experimental $\log_2$FC data for single KO data [39] and triple KO data [38]. Using these values we quantified the agreement between predicted and experimental responses using Spearman correlation (Spearman rank correlation test) and Fraction of Agreement measure (FoA, see methods). IGNITE simulations showed significant correlation with experimental KO–WT differences for Rbpj ($\rho$ = 0.531, $p$ = 9.19 × 10⁻³), Etv5 ($\rho$ = 0.803, $p$ = 3.92 × 10⁻⁶), and triple KO ($\rho$ = 0.716, $p$ = 2.65 × 10⁻⁴) conditions, while Tcf7l1 showed no correlation ($\rho$ = 0.010, $p$ = 9.64 × 10⁻¹), in line with the known post-transcriptional regulation of this factor [43] (Fig 3A and 3B, Table 2).

Moreover, we quantified the agreement between predicted and experimental gene expression changes using the Fraction of Agreement (FoA), defined as the fraction of genes for which predicted and experimental changes occur in the same direction (see Table 3 and Methods). The FoA was 0.65 for Rbpj KO and 0.74 for Etv5 KO, with both values being

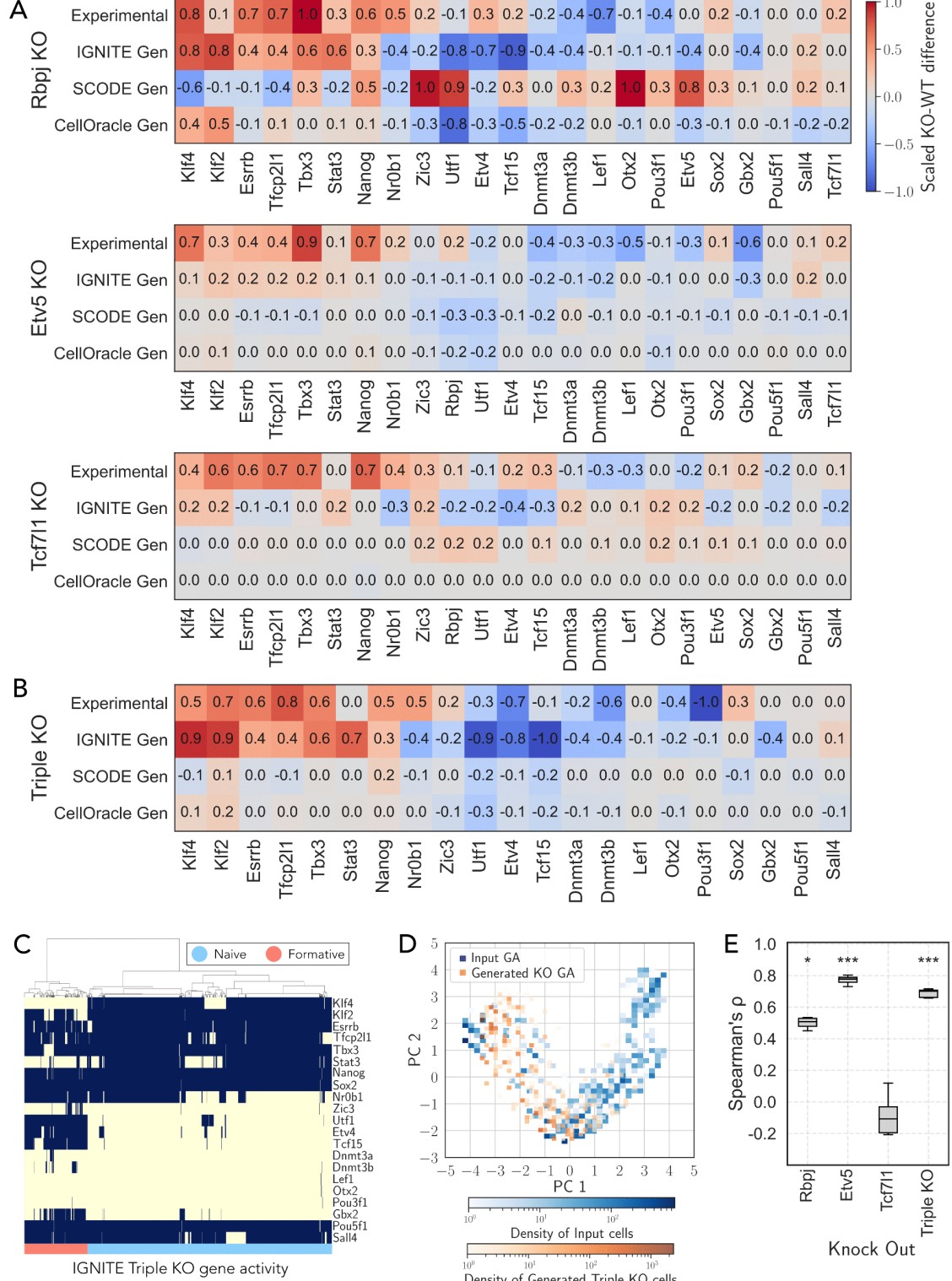

**Fig 3. IGNITE predictions of single and triple KO perturbations in mPSCs and comparison with experimental benchmarks. A.** Scaled KO–WT difference for Rbpj, Etv5, and Tcf7l1 single knockouts, computed from experimental data [39] and from simulations with IGNITE, SCODE, and CellOracle. For each gene, the simulated scaled KO–WT difference was calculated as the scaled difference between the average fraction of active cells in

wild-type and knockout conditions. For the experimental data, scaled log2FC values from [39] were used. All quantities were scaled between −1 and +1 to facilitate comparison across datasets (see Methods for details). **B.** Scaled KO–WT difference for the triple knockout, computed from experimental data [38] and from simulations with IGNITE, SCODE, and CellOracle. The simulated scaled KO–WT difference was calculated as the scaled difference between the average fraction of active cells in wild-type and knockout conditions. For the experimental data, scaled log2FC values from [38] were used. All quantities were scaled between −1 and +1 to enable comparison across datasets (see Methods for details). **C.** Hierarchical clustering of the IGNITE-generated triple KO GA. The clustering algorithm used is Ward's method. Each row represents a gene, while each column corresponds to an individual cell. The color indicates inactive (−1, yellow) or active (+1, blue) gene activity. 9547 cells were simulated. **D.** PCA scatter plot representing the gene activity, GA, of the input dataset (scRNA-seq data with LogNorm, PST, and MB) and the IGNITE-generated triple KO GA. The colour gradient represents the cell density within each square area, with separate scales for the input (blue) and the generated triple KO cells (orange). **E.** Spearman's correlation between simulated and experimental scaled KO–WT differences for Rbpj, Etv5, Tcf7l1, and the triple KO. Results are based on the 10 GRNs inferred with IGNITE that exhibited the lowest CMD values (out of 250 tested models). Boxes indicate the interquartile range, the horizontal line marks the median, and whiskers represent the full range of non-outlier data.

**Table 2. Spearman correlation (Spearman rank correlation test) between predicted and experimental KO–WT average gene activity changes for each method. Values in parentheses indicate the associated *p*-values in scientific notation. Statistically significant correlations (*p* < 0.05), whether positive or negative, are shown in bold.**

| KO | IGNITE | SCODE | CellOracle |
|---|---|---|---|
| Rbpj | **0.53** ($p = 9.19 \times 10^{-3}$) | **−0.46** ($p = 2.56 \times 10^{-2}$) | 0.35 ($p = 1.04 \times 10^{-1}$) |
| Etv5 | **0.80** ($p = 3.92 \times 10^{-6}$) | −0.05 ($p = 8.05 \times 10^{-1}$) | **0.52** ($p = 1.19 \times 10^{-2}$) |
| Tcf7l1 | 0.01 ($p = 9.64 \times 10^{-1}$) | −0.30 ($p = 1.62 \times 10^{-1}$) | **−0.55** ($p = 6.49 \times 10^{-3}$) |
| Triple KO | **0.72** ($p = 2.65 \times 10^{-4}$) | 0.11 ($p = 6.42 \times 10^{-1}$) | **0.57** ($p = 7.45 \times 10^{-3}$) |

statistically significant compared to null models (binomial test; see Methods). In contrast, no significant agreement was observed for Tcf7l1 KO.

The considered triple knockout has been reported to expand for several passages under the experimental conditions promoting differentiation, while retaining expression of naïve markers in the vast majority of cells [38]. As IGNITE could simulate the cellular composition of wild-type differentiating cells (Fig 2E), we asked whether it could simulate the cell composition of triple KO cells (Fig 3C). Most cells (79%) displayed activation of all naïve genes, the remaining 21% exhibited activity of some formative genes (Utf1, Etv4 and Tcf15) and inactivation of only two naïve genes (Klf4, Stat3).

Comparison of the PCA of triple KO gene activity generated with IGNITE and the PCA of the input MB gene activity data (Fig 3D) showed that triple KO simulations occupied regions corresponding to early time points of the input data, characterized by a predominance of naïve-state cells. These observations are consistent with previously reported experimental findings for the triple KO [38].

To evaluate the results of IGNITE, we simulated the single and triple gene perturbation with SCODE and CellOracle [8]. We computed the scaled KO-WT difference for SCODE and CellOracle simulations and compared it with the experimental log2FC (Fig 3A and 3B). Across the tested KO conditions, IGNITE yielded higher Spearman correlation coefficients and FoA values for Rbpj, Etv5, and the triple KO compared to SCODE and CellOracle: for SCODE, Spearman correlations ranged from $\rho = -0.46$ to 0.11, whereas CellOracle correlations ranged from $\rho = 0.35$ to 0.57 (Table 2), and FoA values consistently lower than those obtained with IGNITE (Table 3). Furthermore, SCODE-generated KO data did not exhibit an upregulation of naïve genes compared to the WT condition, with the proportion of naïve cells only slightly increasing from 21% in the WT condition (S2D Fig) to 28% in the KO simulation (S2F Fig). Regarding CellOracle, the WT simulation produced a small population of naïve cells (18%), characterized by high expression of markers such as Klf4, Klf2, and Nanog, which is consistent with the expected composition of differentiating wild-type cells (S2G Fig). However, in the triple KO simulation, CellOracle produced an increased proportion of naïve-like cells (48%) compared to the WT condition, indicating a shift in the expected direction. Yet, this proportion remains lower than what is reported experimentally, where most triple KO cells maintain a naïve identity (S2H Fig; [38]).

**Table 3. Fraction of Agreement (FoA) between predicted and experimental KO–WT gene activity changes for each method. Statistically significant FoA values ($p < 0.05$, binomial test) are shown in bold.**

| KO condition | IGNITE | SCODE | CellOracle |
|---|---|---|---|
| Rbpj | **0.65** | 0.30 | 0.39 |
| Etv5 | **0.74** | 0.22 | 0.26 |
| Tcf7l1 | 0.30 | 0.30 | 0.13 |
| Triple KO | **0.67** | 0.48 | **0.57** |

For the results obtained so far with IGNITE, we selected the effective GRN as the one with the lowest CMD (Fig 1A). As a further analysis, we verified that the IGNITE performances in simulating perturbations in the biological system are stable for any GRNs with low normalized distances. For this reason, we selected the 10 GRNs with the lowest CMD values and computed the Spearman correlation between simulated and experimental KO–WT differences (Fig 3E). These results indicate that the predictions made by IGNITE are robust to variations among top-performing GRN models.

## 2.5 Assessing the accuracy of GRN inference against experimental benchmarks and other inference methods

Next, we analysed the GRN inferred with IGNITE, asking whether it contained gene interactions previously validated using independent techniques. The GRNs were visualised as interaction matrices in which the interaction from gene $j$ (columns) to gene $i$ (rows) is denoted by the corresponding entry $(i,j)$ in the matrix. Notably, positive interactions are prevalent within the naïve group and within the formative group, as previously reported. A predominance of negative interactions is observed between the naïve and formative genes (Fig 4A).

For comparison, we inferred the GRN with three additional methods, SCODE, CellOracle and MaxEnt. The MaxEnt GRN interaction matrix shows positive interactions among naïve genes, and also among some formative genes, as expected (S3A Fig). However, the pattern of negative interactions between naïve and formative groups observed in the IGNITE GRN is not clearly distinguishable. We computed the interaction matrix using also SCODE and CellOracle (S3B and S3C Fig, respectively). The SCODE matrix does not show recognisable interaction patterns, as there are no distinct blocks of positive or negative interactions. The matrix shows fewer strong interactions. CellOracle inferred five GRNs, one for each cluster identified in the input scRNA-seq dataset. These CellOracle GRNs present interaction patterns similar to IGNITE, with positive interactions within the naïve genes and the formative genes. Only the interaction strength changes among the clusters, while the patterns are preserved.

To quantitatively compare the inferred GRNs, we have identified 18 experimentally validated interactions from the literature (Fig 4B) [38,44–49]. We considered an interaction as experimentally validated when a factor has been shown to directly interact with the promoter of target genes, and its genetic inactivation or over-activation results in a significant change in the levels of its direct targets. In Fig 4C, we presented the IGNITE subnetwork, which includes only the genes involved in the known interactions. Its links are the known interactions correctly inferred. Importantly, statistical tests on null models (permutation test, see Methods) confirmed that these interactions could not be attributed to random chance (S4A Fig).

From the known interactions, we calculated the fraction of correctly inferred interactions (FCI) as the fraction of all inferred interactions that have the same sign as the known interactions (Table 4) and Methods for details). We focused exclusively on the signs of the interactions since their strength depends on the model and remains experimentally unmeasurable. Given the undirected nature of the MaxEnt GRN, we did not consider the directionality of the known interactions for this method. The FCI for all methods was comparable, indicating a similar ability to identify interactions among all methods tested.

In addition to the FCI, we computed the Spearman correlation between the interaction strengths in the GRN inferred by IGNITE and those inferred by the other methods, in order to assess the similarity in the overall interaction structure. While

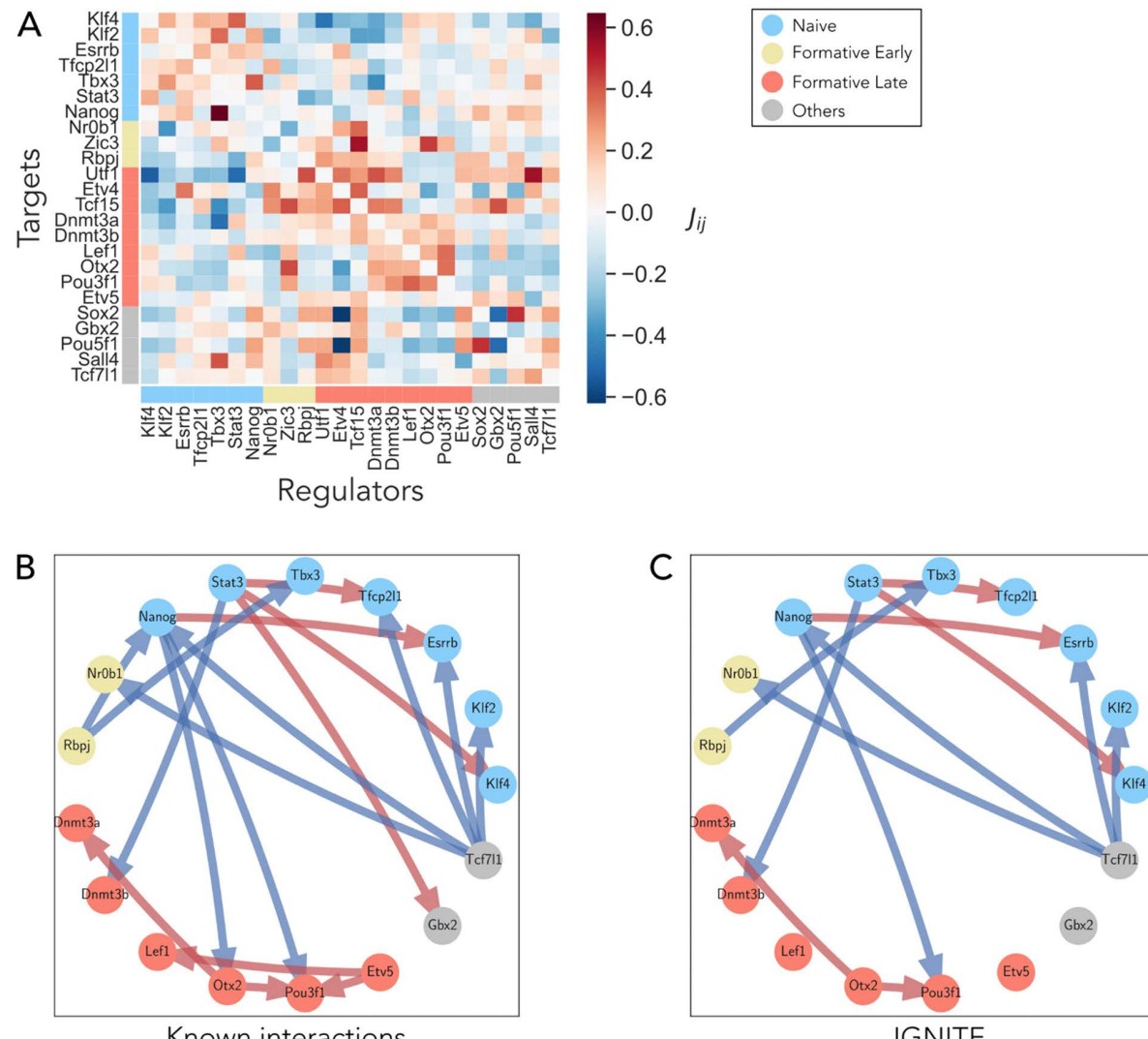

**Fig 4. GRN inferred by IGNITE and comparison with literature-supported interactions. A.** Interaction matrix of the GRN inferred with IGNITE from the input dataset (log-normalized scRNA-seq with PST and MB). Genes are grouped as in Fig 1C: naïve (blue), formative early (yellow), formative late (red), and others (grey). Matrix elements (*i,j*) represent the interaction from regulator *j* to target *i*, with positive (red) and negative (blue) values. **B.** Subnetwork of 18 interactions reported in the literature. Only the involved genes are shown, coloured as in A. Arrows denote activating (red) or inhibitory (blue) interactions. **C.** Subnetwork of literature-reported interactions correctly recovered by IGNITE. The arrows represent the interactions that IGNITE inferred with the correct sign.

MaxEnt and CellOracle showed moderate positive correlations with IGNITE, SCODE exhibited no significant correlation (Table 4).

## 2.6 Evaluating prior knowledge integration and scaling properties

GRN inference, WT generation, validation against known interactions, and KO perturbation analyses were evaluated using a set of metrics for the mPSC dataset (S1 Table). We next use these metrics to assess the impact of prior knowledge integration and scaling properties.

**Table 4. Comparison of the 4 considered inference methods using as input the mouse scRNA-seq dataset (log-normalized and processed with PST and MB). The table reports (i) the fraction of correctly inferred known interactions (FCI) and (ii) the Spearman correlation (with associated *p*-value) between the interaction strengths inferred by IGNITE and each other method.**

| Method | FCI | Correlation (IGNITE) |
|---|---|---|
| IGNITE | 0.67 | – |
| MaxEnt | 0.72 | 0.36 ($p < 2.2 \times 10^{-16}$) |
| SCODE | 0.67 | -0.03 ($p = 5.3 \times 10^{-1}$) |
| CellOracle | 0.67 ± 0.04 | 0.39 ($p < 2.2 \times 10^{-16}$) |

IGNITE takes as input only scRNA-seq data and does not require prior knowledge (Fig 1A), as it selects GRNs by minimizing the CMD. However, other approaches, such as CellOracle, integrate additional experimental datasets (e.g., ATAC-seq) as prior knowledge. We therefore asked whether using experimentally validated interactions as prior knowledge would improve the performance of IGNITE. To do so, we searched for GRNs that maximised FCI. We identified a set of ten GRNs containing 14 of 18 experimentally validated interactions (FCI = 0.78).

To investigate the relationship between the two approaches, we calculated the CMD and FCI for all GRNs generated by IGNITE (Fig 5A). The 10 GRNs maximising the FCI (orange dots) are not the same as minimizing the CMD (red dots) that we analysed in Fig 3E. However, the two parameters anticorrelate (Spearman correlation $\rho = -0.69$, with *p*-value=$5.82e - 36$, Spearman rank correlation test), potentially indicating that the two approaches might identify GRNs with similar properties.

We then assessed the KO prediction performance of these two GRN sets. Table 5 reports the average Spearman correlation ($\rho \pm$ SEM) between simulated and experimental KO-WT differences across the ten GRNs per selection strategy. GRNs selected by minimizing CMD achieved consistent and statistically significant positive correlations for Rbpj, Etv5, and the triple KO, whereas Tcf7l1 was only weakly predicted (see Methods). In contrast, GRNs selected by maximising FCI showed lower and more variable performance, with correlations not significant after FDR correction. These results

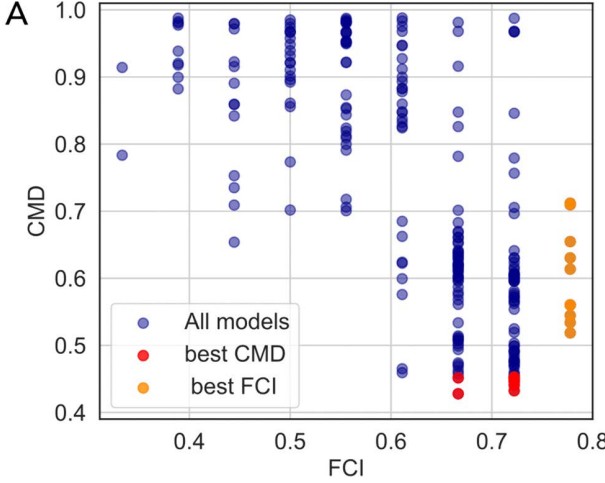

**Fig 5. Comparison of CMD and FCI selection criteria in mouse GRN inference. A.** Relationship between the fraction of correctly inferred interactions (FCI) and the Correlation Matrices Distance (CMD) for the 250 GRNs inferred with IGNITE from the input dataset (log-normalized scRNA-seq with PST and MB). Each point is a GRN, and it corresponds to a specific set of hyperparameters. The 10 models with the lowest CMD are highlighted in red, and the 10 with the highest FCI in orange.

**Table 5. Spearman correlation (mean ± SEM) between simulated and experimental KO–WT differences for the ten GRNs with lowest CMD or highest FCI. P-values are from one-sided t-tests against zero, with FDR correction across the four KO conditions. Significant results are marked with stars (\* $p<0.05$, \*\* $p<0.01$, \*\*\* $p<0.001$).**

| KO | CMD-selected GRNs | | FCI-selected GRNs | |
|---|---|---|---|---|
| | Mean ± SEM | adj. $p$ | Mean ± SEM | adj. $p$ |
| Rbpj | 0.503 ± 0.010 | $2.1 \times 10^{-12}$ \*\*\* | 0.228 ± 0.103 | 0.056 |
| Etv5 | 0.773 ± 0.008 | $2.8 \times 10^{-14}$ \*\*\* | 0.320 ± 0.131 | 0.056 |
| Tcf7l1 | -0.095 ± 0.035 | 0.024 \* | -0.206 ± 0.079 | 0.056 |
| Triple KO | 0.688 ± 0.008 | $3.4 \times 10^{-14}$ \*\*\* | 0.329 ± 0.150 | 0.056 |

indicate that incorporating experimentally validated interactions as prior knowledge does not improve IGNITE inference quality, and CMD remains the most reliable selection criterion.

We next assessed scalability with respect to gene and cell number. We generated with IGNITE GRNs of increasing size from 24 to 200 genes using as input our large scRNA-seq data. We have performed this analysis and measured the computation time on a standard laptop computer (MacBook Pro with an Apple M1 Pro processor and 16GB of RAM). The computational time fits a quadratic trend with GRN size (with a quadratic fit yielding $R^2 = 0.9997$), reaching $\sim$ 12 hours for the largest GRN, which is an acceptable computational time. To evaluate scaling with respect to cells, we randomly subsampled 500 cells 25 times ($\sim$5% of the dataset), retaining the best GRN in each batch. As shown in Table 6, the best GRN inferred from only 500 cells still reproduced experimental KO–WT differences with high accuracy for Rbpj, Etv5, and the triple KO, while Tcf7l1 remained less predictable (see Methods). These results demonstrate that IGNITE maintains KO-prediction fidelity even with limited input data, while also reducing runtime, supporting its practical scalability. Encouraged by this scalability, we next applied IGNITE to an independent dataset of human PSCs to assess generalizability across species and experimental platforms.

## 2.7 Human GRN inference and WT data generation

The human PSCs scRNA-seq dataset from Chu et al. [31] captures the cell differentiation from pluripotency to mesendoderm and finally to definitive endoderm (DE) (Fig 6A). It consists of 758 cells collected at six time points (0, 12, 24, 36, 72, and 96 hours) (S5A Fig). We selected this dataset for three main reasons: i) It is well studied and has been previously analyzed for GRN inference by Matsumoto et al. and Pratapa et al. [10]. ii) It was generated using the Fluidigm C1 system, a single-cell transcriptomics technique different from the 10X platform used in the mouse dataset. This system captures fewer cells, allowing us to assess IGNITE performance with more limited input data. iii) It represents a biological system distinct from the previously analysed mouse model.

The hPSC analysis follows the same benchmarking framework applied to the mPSC dataset, including GRN inference, WT data generation, validation against known interactions, and KO perturbation analysis, using dataset-specific metrics where appropriate (S1 Table). As required by IGNITE, we normalized the data using logarithmic normalization,

**Table 6. Spearman correlation (mean ± SEM) between simulated and experimental KO–WT differences for the best GRN inferred from 500 randomly subsampled cells. P-values from one-sided t-tests against zero with FDR correction across KO conditions.**

| KO | Mean ± SEM | adj. $p$ | Significance |
|---|---|---|---|
| Rbpj | 0.554 ± 0.011 | $<10^{-12}$ | \*\*\* |
| Etv5 | 0.678 ± 0.021 | $<10^{-12}$ | \*\*\* |
| Tcf7l1 | 0.273 ± 0.078 | $9.0 \times 10^{-4}$ | \*\*\* |
| Triple KO | 0.697 ± 0.008 | $<10^{-12}$ | \*\*\* |

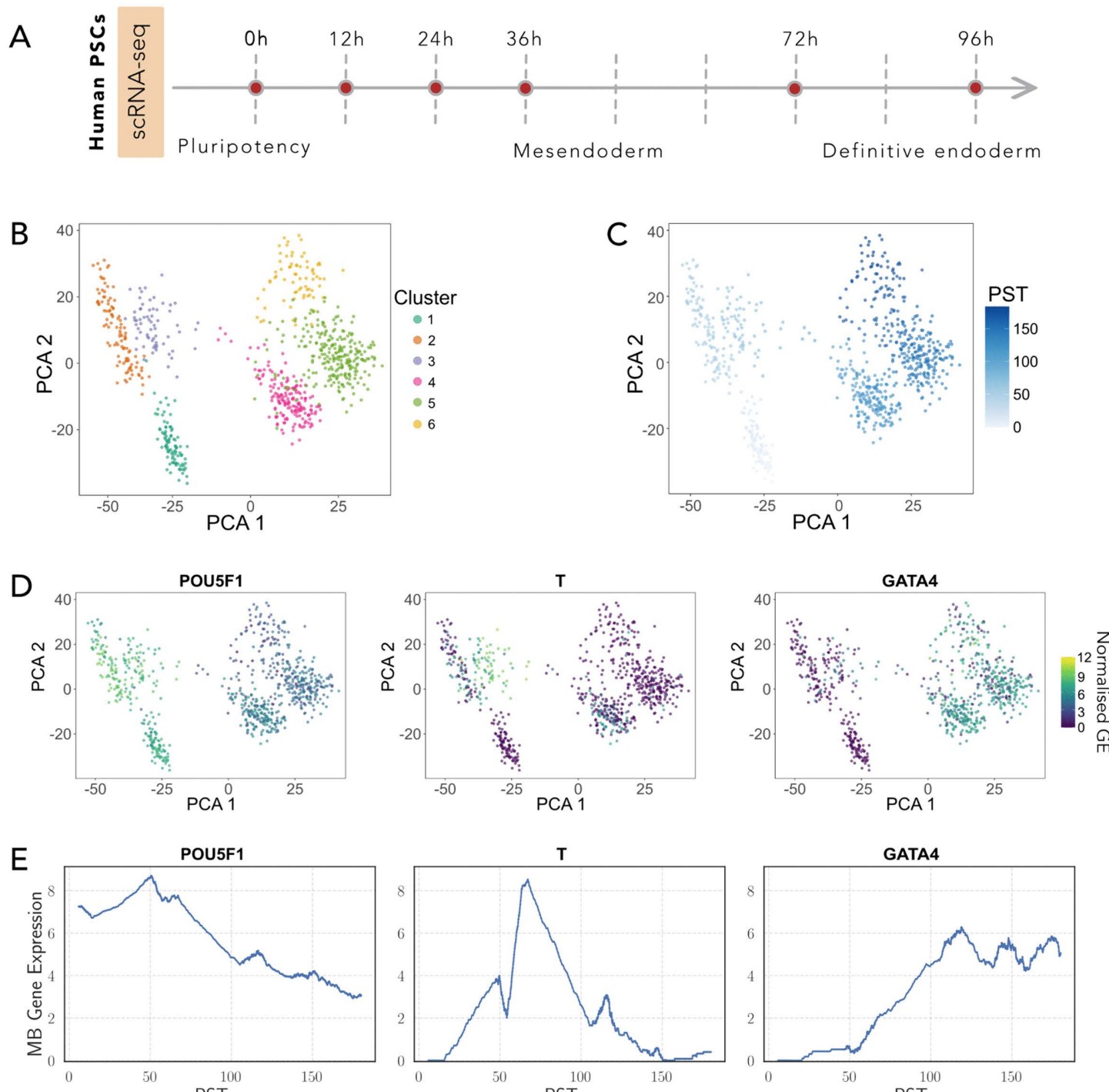

**Fig 6. Human PSC dataset preprocessing. A.** Studied biological system [31]: temporal dynamics of human PSC differentiation, spanning pluripotency, mesendoderm, and definitive endoderm. The dataset is based on Smart-seq scRNA-seq. Cells were sampled at six time points: 0h ($n=92$), 12h ($n=102$), 24h ($n=66$), 36h ($n=172$), 72h ($n=138$), and 96h ($n=188$). **B.** PCA of the log-normalized dataset. Each point represents a cell, coloured by clustering label. **C.** PCA of the dataset, with cells coloured by pseudotime. The pseudotime ranges from the value of 0 to the final value of 183.58. **D.** PCA visualisation of three key genes (Pou5f1, T, and Gata4) in the log-normalized scRNA-seq dataset. Each point represents an individual cell, coloured by the normalized expression of the respective gene. **E.** Mini-Bulk gene expression patterns in pseudotime for three representative genes, Pou5f1, T, and Gata4.

projected the data into a lower-dimensional PCA space and clustered the cells to compute the pseudotime (Fig 6B, see Methods for details).

We computed PST using the Slingshot algorithm [32]: the PST values aligned with experimental time points (Fig 6C), and the distribution of PST within each sampling time confirmed the expected ordering (S5B Fig). To assess the biological coherence of the identified clusters, we examined the cluster composition across time (S5C Fig) and the expression distribution of representative genes within clusters (S5D Fig), which matched the trends reported by Chu et al. [31]. Key temporal dynamics were preserved after PST ordering (Fig 6D and 6E): cells at low PST showed higher expression of pluripotency markers (e.g., POU5F1), mesendoderm markers peaked at intermediate PST (e.g., T), and DE markers increased at higher PST (e.g., GATA4).

As a next step, we defined the GRN nodes. We selected all genes annotated as transcription factors (TFs) [50] present in the dataset, retaining those with significant differential expression in at least one cluster. This selection resulted in a set of 92 genes forming the basis for GRN inference (see Methods), plus 6 genes used for validation. We then constructed the Mini-Bulk profiles by aggregating neighboring cells along PST. The resulting heatmap (S5E Fig) highlights distinct temporal patterns: a few genes peak early, whereas most are activated at intermediate or late stages. Finally, we binarized expression to obtain the Gene Activity dataset used for GRN inference (S5F Fig).

## 2.8 Evaluating human GRN against known interactions

Using as input only GA, the processed scRNA-seq data, IGNITE inferred a set of 250 gene regulatory networks (GRNs) and computed the Correlation Matrix Distance (CMD) for each (S6A Fig). We selected the GRN that best reproduced the correlations among genes (Fig 7A and 7B), with a CMD of 0.35. Statistical validation using a two-sample t-test confirmed a significant difference between the CMD values generated by the IGNITE model and those from the null model (t-statistic: $t = -66.01$, p-value: $p < 2.2 \times 10^{-16}$). The inferred GRN is represented as an interaction matrix, where each entry $(i,j)$ represents the interaction strength from gene $j$ (columns) to gene $i$ (rows) (S6B Fig). The interaction matrix reveals a heterogeneous pattern of interactions with positive and negative regulatory relations, with potential regulatory modules and interaction patterns. We selected interactions among 22 genes with a known role in differentiation of human PSCs to definitive endoderm (S6C Fig) [31,51]. The interaction matrix reveals a predominance of positive interactions within groups of genes active in different stages of differentiation. In contrast, negative interactions are predominantly observed between genes expressed at different stages of differentiation. Specifically, naïve pluripotency genes tend to repress mesendoderm markers, such as T inhibition and definitive endoderm genes, e.g., HNF4A and SOX17 inhibition. Conversely, late-stage genes negatively regulate early-expressed factors, e.g., the repression of NANOG and POU5F1.

To validate the inferred GRN, we selected genes that are part of the TRRUST regulatory network, which was previously used by SCODE for validation [52]. IGNITE successfully inferred 6 out of the 7 interactions shared between our GRN and the TRRUST network (S6D Fig).

Furthermore, IGNITE generated 150 gene activity datasets with the same dimension of the input dataset. We assessed the quality of IGNITE-generated data by comparing the input GA with the IGNITE-generated WT GA in the PCA space. Each binary dataset generated by IGNITE was projected into the PCA space of the WT gene activity data (Fig 7C). To quantitatively evaluate the dataset similarity, we defined the regions corresponding to the previously identified clusters in this PCA space by using input WT data (see Methods). Cluster 5 and Cluster 6 were merged, as the region of Cluster 6 was entirely contained within that of Cluster 5 in the PCA computed from GA data. This result was expected, as these clusters correspond to cells with higher PST values (S6E Fig), which in turn correspond to later time points (72 and 96 hours), where overlapping domains between single cells had already been observed in Chu et al. [31]. By leveraging the defined cluster areas, we verified that the IGNITE-generated cells reproduced the same patterns of activation of the input cells without introducing cell types that were not present in the original dataset. Furthermore, we confirmed that IGNITE

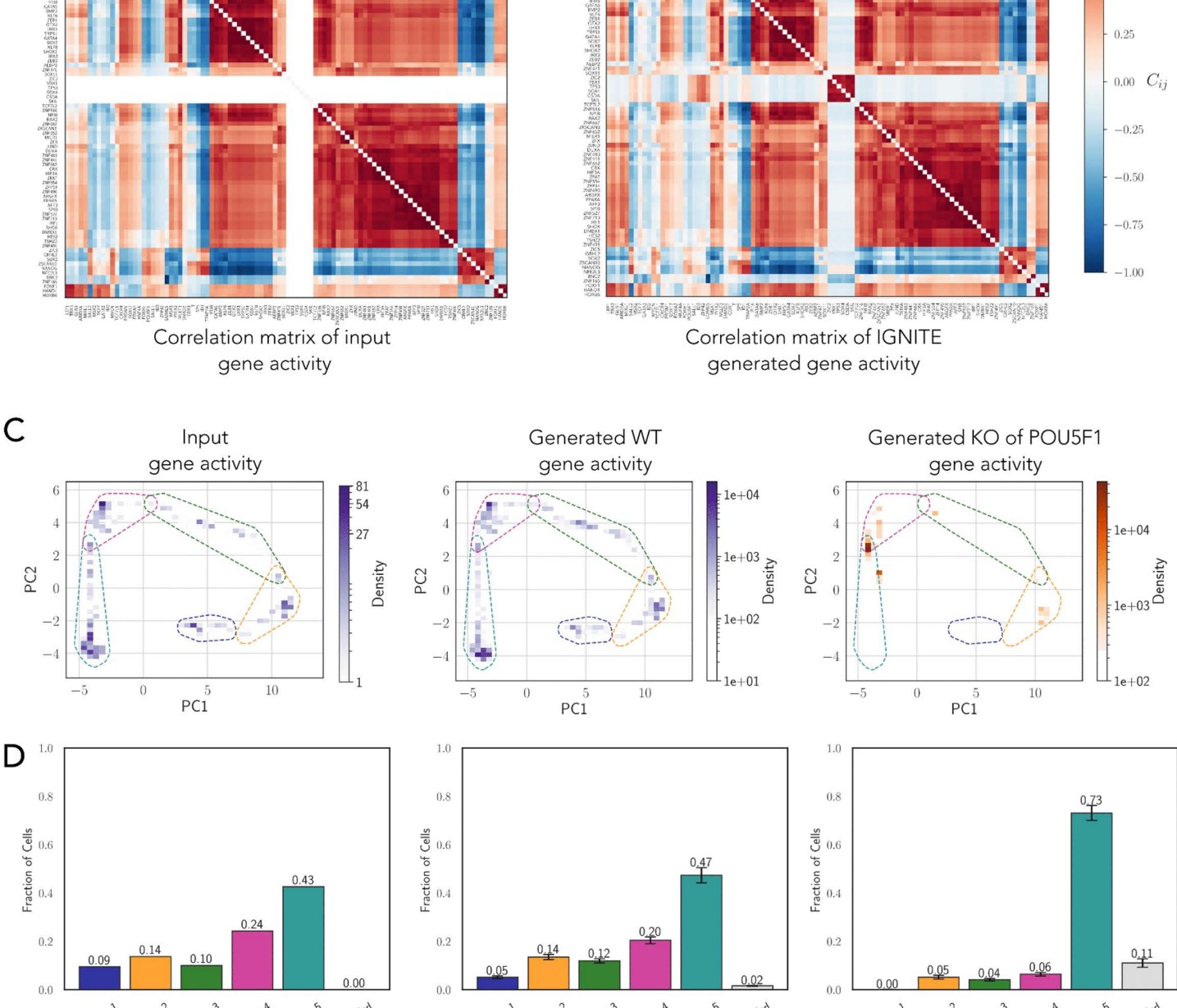

**Fig 7. Inference and validation of IGNITE in hPSCs: GRN and generated data. A.** Pearson correlation matrix of gene activity in the input dataset (human PSCs differentiation scRNA-seq data with LogNorm, PST, and MB). **B.** Pearson Correlation matrix of the gene activity generated with IGNITE. **C.** PCA of gene activity for three datasets: input, from scRNA-seq data with LogNorm, PST, and MB (left), IGNITE-generated wild type (middle), and IGNITE-generated knockout of *POU5F1* (right). The colour gradient represents the cell density within each square area. Dashed contours denote cluster boundaries defined on the input (left) and reused for the generated datasets. **D.** Fraction of cells assigned to clusters (1–5) or excluded areas, for the same three datasets shown in C. For generated data, bars show means across replicated simulations (150 simulated datasets; error bars indicate SEM). Bar colours match the clusters in C.

accurately reproduces the fraction of cells in each cluster, as shown in Fig 7D, correctly inferring the cells' pattern of compositions in the dataset in WT state (see Methods).

### 2.9 Human PSC KO predictions

To assess the ability of IGNITE to predict cell pattern changes following gene knockouts in human PSCs, we generated gene activity profiles under knockout conditions for each gene in the GRN. The effects of gene KO are reflected in variations in gene activity patterns between WT and KO conditions. To assess these changes, we examined the distribution of KO-generated cells in PCA space (Fig 8A) and analyzed how the fraction of cells in each cluster differed from the WT condition (Fig 8B).

To validate these KO perturbation effects, we compared IGNITE simulations with the analysis by Chu et al. [31]. The expected behavior observed in such analysis is that T expression transiently peaks early in differentiation, while CXCR4 is upregulated at later stages, marking the transition from mesendoderm to DE. Chu et al. defined a Differentiation Score (DS) as the ratio between the percentage of CXCR4$^+$ cells and the percentage of T$^+$ (EGFP$^+$) cells, measured by flow cytometry a day 2 of differentation in the KO condition, normalized by the same ratio in WT (see Methods for details). A score lower than 1 indicates impaired differentiation, while a score greater than 1 suggests an improvement in the process.

To compare IGNITE predictions with Chu et al., we computed the fraction of cells expressing CXCR4 and T in each cluster, both in the WT condition (S7A Fig) and under the KO condition for the selected genes. In the WT condition, cluster 4 exhibits a decrease in T$^+$ cells and a concurrent increase in CXCR4 cells, mirroring the trend observed at day 2 of differentiation in the experimental data. Thus, we then computed the DS as the ratio between the fraction of CXCR4$^+$ and T$^+$ cells in cluster 4 under KO condition, normalized by the same ratio in the WT condition. This yielded the DSs for the various KO simulations presented in Fig 8C.

Consistent with the experimental data from Chu et al., only POU5F1 KO enhances differentiation, as indicated by a DS greater than 1. In contrast, SOX17 KO resulted in a score close to 1, suggesting no significant effect on differentiation, as expected. All other KOs impaired differentiation, with KLF8 KO displaying the lowest DS. This results indicate that IGNITE correctly predicted the general impact of gene KOs on differentiation.

Furthermore, we systematically analyzed the effects of KO perturbations for all components of the GRN. We computed, for each simulated KO, the difference in cell fractions across clusters between WT and KO conditions. A positive difference indicates an increase in the proportion of cells within a given cluster under the KO condition, while a negative difference indicates a decrease.

We performed hierarchical clustering of the fraction of cells in each cluster for each condition (either gene KO or WT), obtaining 11 different behaviors. Groups of KOs enhancing differentiation include Group 1 (that contains, for example, POU5F1) and Group 7, where knockouts lead to an increased fraction of cells in later clusters (Cluster 5 and Cluster 4, respectively). This suggests that these genes normally act as differentiation inhibitors, and their loss facilitates progression toward more mature cell states. This effect is particularly evident for POU5F1, whose KO is known to drive cells toward later developmental stages, as previously discussed. Conversely, Groups 3, 4, 5, and 6 contain KOs that impair differentiation (e.g., EOMEs, ID1, ID2), leading to an accumulation of cells in earlier clusters (Clusters 1, 2, or 3). Groups 8 and 9 (e.g., KLF8, CXCR4, T) show inhibition of later-stage differentiation, with cells accumulating in mid-stage clusters rather than being completely arrested in the earliest differentiation states. In this cluster, there are also the outcomes for the KO of NANOG and SOX2, which we already discussed. Finally, Groups 10 and 11 (e.g., SOX17) comprise genes whose KO reduces the proportion of cells in later clusters (4 and 5) while slightly increasing the presence of cells in earlier clusters. This suggests that these genes play a crucial role in stabilising the definitive endoderm fate.

We then compared our analyses with experimental data for known regulators of these developmental stages. The KO of POU5F1 enhances differentiation, decreasing the proportion of cells in the early clusters while increasing those in

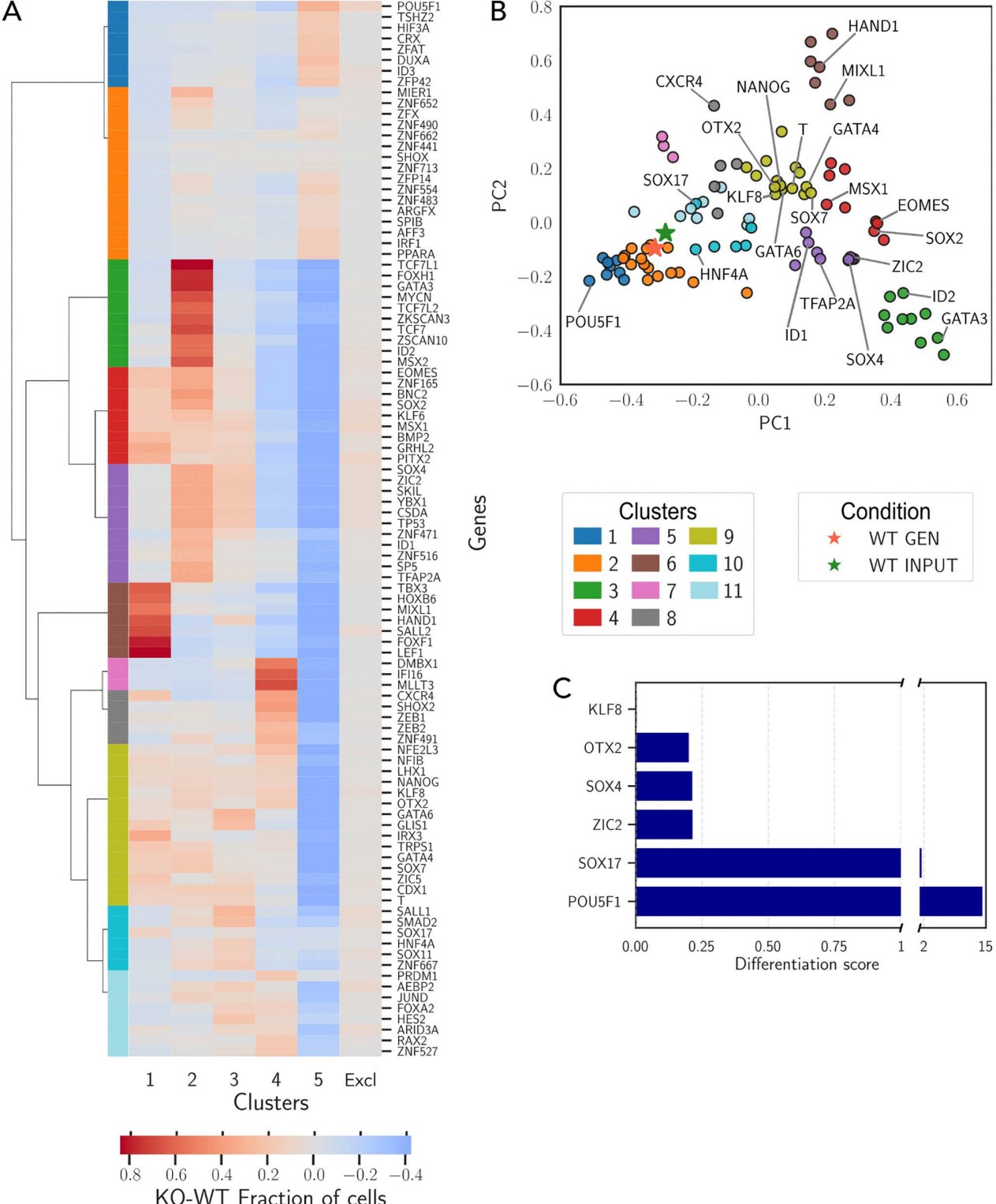

**Fig 8. Evaluating IGNITE predictions of gene knockouts in hPSC differentiation. A.** Difference in cluster composition between WT- and KO-generated data, averaged across 150 replicates. Rows correspond to simulated single-gene KOs, columns to clusters (1–5) or excluded cells. Rows were ordered by hierarchical clustering, which also identified 11 groups shown by colour. Red indicates an increase after KO, blue a decrease. **B.** PCA of

cluster composition across conditions: WT input, WT-generated, and each single-gene KO. Each point corresponds to one condition and is positioned by the fractions of cells across clusters (1–5) and excluded cells (fractions for generated conditions are averaged over 150 replicates). Colours indicate the groups defined in Fig 8A. KO conditions corresponding to the genes shown in A are indicated. **C.** Differentiation score for selected KO genes, chosen to overlap with those analysed in [31], computed as described in Methods. The axis break highlights the outlier magnitude of *POU5F1*.

Cluster 5. This result is consistent with experimental observations, where POU5F1 KO has been shown to cause a loss of pluripotency [31] and accelerated differentiation. We further inspected experimental data from an indepdent study [51] observing a reduction in pluripotency genes and increased expression of endoderm genes (S7B Fig). KO of NANOG caused an increase in the fraction of early differentiation clusters (S7C Fig). This is in line with data from a previous study [51] indicating increased expression of early endoderm markers. The KO of genes that are positive inducer of the endoderm fate (e.g., HNF4A, CXCR4, KLF8 and SOX17), resulted in an increased presence of cells in the early clusters, corresponding to the initial differentiation stages, and a decrease in Clusters 4 and 5, associated with later differentiation stages. This is in line with impaired endoderm differentiation, as reported in previous studies [31,51]. Overall, IGNITE correctly inferred the differentiation effects of KOs, predicting differentiation-promoting and differentiation-blocking gene KOs.

## 3 Discussion

In silico approaches can be applied to modelling and predicting the effects of perturbations on cells, such as knockouts, forced gene activation, or external stimuli. An efficient algorithm would allow to inspect a large number of perturbations on a system, prioritising those most likely to have an effect upon experimental validation. Many methods focused on GRN inference [10] or on predicting perturbation effects [7,8]. However, previous perturbation methods required model training on perturbed data, limiting their scope [7]. Furthermore, a variety of algorithms rely on different types of data as input, downplaying their applicability depending on data availability and compatibility. For instance, algorithms such as RE:IN [11] require ChIP-Seq and bulk transcriptomics data, and need knockout experiments to build the GRN. Understanding how to link cell phenotypes with inferred GRN and how different conditions affect cell behavior remains challenging.

To address this fundamental biological question, we developed IGNITE, a method that exploits GRN inference to simulate gene perturbations. The significant advantage of IGNITE is its ability to inspect the role of gene interactions in shaping cell phenotypes. We demonstrated its capabilities on two complementary systems of pluripotent stem cells (PSCs): mouse PSCs transitioning from the naïve to formative state, and human PSCs differentiating toward definitive endoderm [31]. These datasets differ in species, developmental trajectory, and single-cell platform (10X vs. Fluidigm C1), providing a stringent test of generalizability. IGNITE offers several advantages. (1) It is capable of generating wild-type data, replicating the gene activation patterns within each cell, and (2) simulating GRN perturbations, such as gene knockouts, as well as accurately forecasting their results, predicting how KOs alter gene activity within the GRN. We verified that the patterns of generated behavior are consistent with the experimental predictions. This predictive capability paves the way for a deeper understanding of the underlying biological mechanisms governing gene interactions. Although we did not explicitly simulate forced gene activations, IGNITE could be extended to incorporate such perturbations. (3) IGNITE starts from unperturbed single-cell RNA sequencing data and thus does not need additional inputs (e.g., ATAC-seq data). (4) Through an unsupervised machine learning approach, IGNITE infers directed, weighted, and signed GRN that best describes observed cell phenotypes, where patterns of activation among gene groups are discernible. The inferred GRN is *effective*, as it does not describe biological interactions among genes in real cells. Rather, it captures the functional effective organisation in GRN linked to specific cell behaviors. These advantages were exemplified in the application of IGNITE to differentiating PSCs, where it inferred effective GRNs that successfully forecast cell phenotypes under wild-type or perturbation conditions, elucidating the behavior of PSCs during differentiation.

To benchmark IGNITE, we adapted and tested other algorithms to produce comparable outputs. For our differentiation data, MaxEnt [16,17] was not sufficiently informative as it can not infer the directionality of the network. SCODE [18],

designed for GRN inference from scRNA-seq data of differentiating cells, does not model the intrinsic variability seen in the data, as it performs inference with a deterministic model. Furthermore, SCODE does not exploit the model's ability to generate simulations of system perturbations, even when feasible. We explored this untested SCODE generation capability to simulate single and triple KOs. CellOracle [8] uses GRNs inferred from single-cell multi-omics data to perform in silico perturbations of transcription factors, simulating the resulting changes in cell identity based on unperturbed wild-type data. However, this approach does not describe the underlying process, as it focuses on the propagation of the perturbations in the GRN. We did not employ CellOracle to dissect cell identity via network inference. Rather, we limited its application to measure the effects of KO perturbations, in a way comparable to IGNITE. We included the triple KO simulation, a scenario that was not investigated in the original work. Even though all methods were able to correctly infer known interactions in the GRN, IGNITE outperformed all of them in generating both wild-type and perturbed data. This is because IGNITE leverages its generative non-linear dynamics to learn the properties of the underlying effective GRN, rather than relying on data-constrained GRN inference or perturbation rules to propagate gene expression changes after KOs.

To further evaluate the robustness and generalizability of IGNITE, and to test whether the benchmarking framework established for mPSCs holds across species and experimental platforms, we applied it to a previously published scRNA-seq dataset capturing the differentiation of human PSCs [31]. This dataset differs from the original mouse model biologically and technically: it involves human cells undergoing a different developmental trajectory and was generated using the Fluidigm C1 platform, which captures fewer cells (758) than the 10X Genomics platform (9894). Despite the reduced number of cells and the lack of prior knowledge for gene selection, IGNITE successfully inferred a biologically meaningful gene regulatory network composed by 98 genes selected unbiasedly from scRNA-seq data as the most variable TFs. The inferred GRN revealed coherent regulatory patterns, including stage-specific positive and negative interactions between gene groups, aligning with known differentiation dynamics. Furthermore, IGNITE effectively reproduced the wild-type gene activity distribution, capturing both the global structure of the differentiation process and the relative abundance of cells across clusters. Notably, IGNITE predicted KO perturbations resulting in blockade of differentiation at different stages. Some KOs were predicted to enrich the population of pluripotent cells, while others enrich for pre-endoderm cells, suggesting that different genes promote differentiation at different stages of the process. IGNITE correctly predicted the phenotypic effects of gene knockouts, in agreement with two indipendent experimental studies [31,51]. Together, the mPSC and hPSC analyses demonstrate IGNITE can be effectively applied across diverse single-cell datasets, from two different species, even when prior biological information is scarce or the experimental design differs substantially. This flexibility is essential for studying less-characterized systems, where perturbation experiments are not feasible and curated knowledge is limited.

The versatility of IGNITE notwithstanding, its performance depends on key modelling assumptions that set boundaries to its use. First, IGNITE requires a correct pseudotime ordering. This is a crucial step shared by other inference algorithms, such as SCODE, as temporal patterns of gene expression remained consistent after PST ordering. PST and MB allow us to avoid alternative methods based on assumptions that might not precisely capture the actual biological variability, such as the imputation of zero gene expression values. Second, IGNITE adopts an Ising model that simplifies gene expression into binary states by approximating genes as spins. This abstraction has proven sufficient to capture the essential regulatory logic and to predict the effects of perturbations. Moreover, despite operating under additional simplifying assumptions—such as modelling only stationary trajectories via a non-equilibrium steady state [19], assuming a static network, and restricting interactions to pairwise forms—IGNITE was able to reproduce all observed gene activity patterns in the input data. Nevertheless, the framework is naturally extensible: for instance, adopting multistate variables, as in the Potts model [53], could allow the representation of more complex expression dynamics, and relaxing other assumptions could enable the modelling of time-varying or higher-order regulatory interactions. Third, IGNITE uses scRNA-seq data, which may not allow the inference of specific types of gene interactions. For instance, the effects of a Tcf7l1 KO could not be precisely predicted. This outcome was expected since TCF7L1 is regulated at the level of protein stability and DNA

affinity. These aspects cannot be captured by scRNA-seq. The inclusion of other prior knowledge into the model is possible, although we have shown that it does not appear to enhance its performance. This suggests that adding predefined constraints may not always improve performance and could potentially lead to suboptimal solutions. Indeed, the model only provides an effective description of the GRN, and integrating existing knowledge may not always be straightforward or advantageous. This highlights the importance of a cautious approach when integrating known information into such algorithms. Our results are promising since we have found IGNITE to be stable and robust in searching for optimal solutions. This stability is crucial because it implies that IGNITE is capable of finding truly informative GRNs with biological meaning.

Overall, IGNITE is a valuable machine learning approach for studying PSC differentiation. Its applicability can be generalised to characterise different cell systems and their GRNs without requiring experimental perturbations, as several other models [54]. On the contrary, not only IGNITE relies on unperturbed data, but it also enables reliable in silico predictions on system perturbations that can be implemented even for genes with an unknown binding motif. By inferring regulatory interactions even for genes with low expression levels, it also addresses the issue of data sparsity inherent in scRNA-seq datasets. This makes IGNITE a powerful tool for navigating the complexities of gene regulatory networks, even in the context of sparse or low-expression data. Furthermore, IGNITE provides a powerful and scalable framework for uncovering how gene interactions shape cell patterns from single-cell data, offering a path toward predictive modelling of cell fate decisions in complex biological systems.

## 4 Methods

### 4.1 Mouse scRNA-seq data processing and analysis

**4.1.1 Experimental procedure. Maintenance of mESCs:** Mouse ESCs (R1) were cultured as previously described [55]. Briefly, the base medium is HyClone DMEM/F12 base medium without Hepes (Cytiva), with 4mg/mL AlbuMAX II Lipid-Rich Bovine Serum Albumin (GIBCO), 1× MACS NeuroBrew-21 with Vitamin A (Miltenyi Biotec), 1× MEM NEAA (GIBCO), 50U/mL Penicillin–Streptomycin (GIBCO), 1mM Sodium Pyruvate (GIBCO), and 1×2-Mercaptoethanol (GIBCO). The base medium was supplemented with 3.3mM CHIR-99021 (Selleckchem), 0.8mM PD0325901 (Selleckchem), and 10ng/mL hLIF (provided by the VBCF Protein Technologies Facility) to create the 2iL self-renewal medium. Cells were cultured on Greiner Bio-One CELLSTAR Polystyrene 6-well Cell Culture Multiwell Plates, pre-coated with Poly-L-ornithine hydrobromide (6 $\mu$g/mL in 1xPBS, Sigma-Aldrich, for 1 hour at 37°C, SigmaAldrich) followed by Laminin from Engelbreth-Holm-Swarm murine sarcoma basement membrane (1.2 mg/mL in 1xPBS for 1 hour at 37°C, SigmaAldrich). Routine cells passaging was carried out at a 1:6 ratio every 2 days using 1× Trypsin–EDTA solution (Sigma) at 37°C, and the reaction was stopped by using 10% fetal bovine serum (FBS, Sigma) in the base medium.

**Mouse 10X Genomics scRNA-seq data with MULTI-Seq Barcoding and Validation:** 10X Genomics scRNA-seq was performed as described in [55]. Cells were harvested via trypsinization for 10 mins at 37 °C. Trypsinization was stopped using 10% FBS in base medium. Subsequently, the cell pellets were resuspended in PBS to remove any remaining FCS, and their quantification was carried out using a CASY cell counter (Biovendis). Cell labelling with MULTI-Seq barcodes followed a protocol as described in [56]: 0.5 million cells per sample were resuspended in PBS and incubated with a mixture of barcodes and lipid anchors for 5 minutes on ice. Co-anchors were subsequently introduced for an additional 5 minutes, followed by quenching the reaction with 1% BSA/PBS. Cell washing was performed twice using 1% BSA/PBS, and then cells were resuspended in 0.04% BSA. After combining all samples, filtration was carried out through FACS strainer cap tubes and the concentration was adjusted to 1 million cells/mL. Following the manufacturer's instructions, only pools exhibiting > 80% cell viability were used for the subsequent preparation of the 10X library. For MULTI-Seq identification, libraries were prepared in parallel with cDNA libraries sequenced on Illumina NextSeq550 or NovaSeq platforms. To confirm the specificity of MULTI-Seq, ATTO-488 and ATTO-590 conjugated barcodes were used, as detailed in the [56]. Cell labelling with barcodes followed the aforementioned procedure, with label validation performed through FACS analysis.

**4.1.2 The mouse scRNA-seq dataset.** cDNA reads were mapped to the mm10 reference genome and demultiplexed into droplets by using Cellranger as described in [55]. MULTISeq barcodes were mapped to droplets and counted using CITE-Seq-Count and the Seurat vignette "Demultiplexing with Hashtag oligos" was used to perform cell classification. The cells in the dataset belong to 5 time points (0, 6, 12, 24 and 48 h).

**4.1.3 Quality control and cell filtering.** To ensure data quality, we removed doublet and negative cells from the analysis. Per-cell QC metrics were computed using the `perCellQCMetrics` function from the scater R package. Outlier cells were identified and removed using the `quickPerCellQC` R function with a threshold of 3 median absolute deviations (MAD) from the median. In addition, genes with low expression levels (observed in fewer than 20 cells) were excluded from the analysis. After implementing these steps, the dataset comprised $N_{genes} = 13833$ genes and $N_{cells} = 9894$ individual cells.

**4.1.4 Normalization.** We normalized the raw gene expression counts by dividing the total counts for each cell by the corresponding size factor (the sum of the counts for that cell). Subsequently, we log-transformed the normalized expression values by adding *PseudoCount* = 1 and performing a log2-transformation. Therefore, indicating the raw gene expression for the gene $i$ in the cell $\alpha$ as $x_i^\alpha$, we get its normalized gene expression $\bar{x}_i^\alpha$ as follows:

$$\bar{x}_i^\alpha = \log_2 \left( \frac{x_i^\alpha}{\sum_{j=1}^{N_{genes}} x_j^\alpha} + 1 \right).$$

(1)

**4.1.5 Dimensionality reduction.** Dimensionality reduction was performed to enable clustering and pseudotime inference while preserving the temporal structure of the differentiation process. For the mouse PSC dataset, all dimensionality reduction steps were performed using the same input feature set: a subset of 2318 genes previously established as relevant to this differentiation process [14], identified by hierarchical clustering based on Pearson correlation as described by Carbognin et al.

We initially evaluated Principal Component Analysis (PCA) on this gene set. PCA captured a limited fraction of the total variance and did not clearly separate cells sampled at different time points (S1A and S1B Fig), consistent with the dense sampling and approximately linear nature of the differentiation trajectory. Therefore, we adopted a nonlinear dimensionality reduction strategy based on t-SNE [40] followed by UMAP [41], which more effectively captured local neighborhood relationships and temporal ordering of cells in this dataset. Specifically, we computed a two-dimensional t-SNE embedding and subsequently projected it into a two-dimensional UMAP space (S1C Fig). The UMAP analysis revealed a coherent temporal progression of cells from the cells sample at $t = 0$ $h$ to the sample at $t = 48$ $h$ along the first dimension of UMAP. This progression was discernible, with distinct segregation of cells from different time points, forming an ordered sequence from left to right. This observation was further supported by examining the gene expression patterns of our 24 selected genes within the UMAP space (e.g., Fig 1E).

**4.1.6 Clustering.** We performed a clustering analysis using a graph-based approach on UMAP-transformed data. Initially, we constructed a shared nearest neighbor graph with a total of 30 nearest neighbors ($k = 30$). We used the buildSNNGraph function of the R package scran. Next, we applied the Walktrap algorithm to compute the clusters, implemented in the igraph R package. We merged smaller clusters to form larger ones, drawing on time sample labels to understand clusters composed of cells that were generated at the same time point. This procedure was carried out to reinforce both the robustness and interpretability of our analysis on the evolution of cell subpopulations. To validate the biological significance of the resulting clusters, we investigated the distribution of gene expression of the 24 relevant genes within each cluster (e.g., three genes in S1E Fig). Our exploration uncovered distinct patterns: naïve genes exhibited heightened activity mainly in the left clusters (clusters labelled as 2 and 3), while formative genes demonstrated increased activity predominantly in the clusters in the right part of the UMAP space (clusters 4, 5, and 6).

**4.1.7 2Cell-Like cells cluster.** We investigated the cluster 1, as for the 198 cells in this cluster, the gene expression of the 24 selected genes did not allow us to understand whether it was a cluster with a prevalence of naïve, formative early or formative late genes. We computed the top 100 most up-regulated genes and the top 100 most down-regulated genes in this cluster concerning all other clusters by using the scran R package function findMarkers. Remarkably, we observed that the up-regulated gene set contained genes associated with the 2Cell-Like stage [42]. This finding allowed us to confidently assign an identity to the cells within this cluster as 2CLCs. From this point on, we excluded this cluster from our studied dataset since we are not interested in this cell state.

**4.1.8 Pseudotime.** We employed the Slingshot algorithm [32] to compute the pseudotime (PST) of cells. This algorithm starts from a low-dimensional space and needs cluster labels for each cell. For this purpose, we utilised the UMAP space data, as well as the clustering labels. We set the input of the slingshot function: we indicated cluster "2" as the start cluster and cluster "6" as the end cluster. This choice was made after considering the high gene expression regions in the UMAP space for each of the 24 selected genes. Moreover, we observed that the real temporal progression of the process described by the sampling times and the PST progression are in agreement by generating the violin plots of the pseudotime values for the cells in each time point separately (S1F Fig).

**4.1.9 Gene selection.** The dataset after the pseudotime implementation includes 13833 genes and 9696 cells. We focused only on 24 genes known from the literature as relevant and informative in the naïve to formative transition of PSCs [11,14,38,39].

**4.1.10 Mini-Bulk.** We called Mini-Bulk (MB) the approach we chose to compute the moving average over the cells in the dataset: we averaged subsets of cells with a window size of 150 cells and a step size of one. The choice of this value for the step size ensures that the dimensions of the resulting dataset are comparable to those of the original one. We applied the MB procedure to our single-cell dataset, normalized and with cells ordered by following the pseudotime values. In the end, we obtained a dataset of 24 genes and 9547 cells (S1G Fig). The dataset thus obtained will be the input for all GRN construction methods, except for CellOracle, which does not require PST and MB.

## 4.2 Human scRNA-seq data processing and analysis

**4.2.1 QC and normalization.** The human scRNA-Seq time course dataset (0, 12, 24, 36, 72, and 96 h) from Chu et al. [31] consists of 758 cells derived from the differentiation of human ES cells into definitive endoderm (DE). Cells were pre-filtered by the original authors, retaining those with at least 5000 genes with $TPM > 1$, and expression values were already normalized using median-by-ratio normalization. As an additional quality control step, we removed 4348 genes expressed in fewer than 20 cells. Then, we applied a log-transformation.

**4.2.2 Dimensionality reduction.** To reduce the dimensionality of the human scRNA-seq dataset, we identified the top 200 highly variable genes (HVGs) using the modelGeneVar function from the scran R package. Principal component analysis was then performed on the log-transformed expression values of these HVGs, retaining the first 10 principal components. The PCA coordinates in S5A Fig exhibited a clear temporal structure, indicating that the main axes of variation capture biologically meaningful progression across the differentiation time course.

**4.2.3 Clustering.** We used a graph-based approach for clustering on PCA-reduced data. Initially, a shared nearest neighbor (SNN) graph was constructed with the buildSNNGraph function from the scran R package, setting the number of neighbors to $k = 35$. Clusters were identified using the Walktrap algorithm implemented in the igraph package. The correspondence between cluster labels and time points is shown in S5C Fig. The resulting structure reflects a clear temporal progression, with specific clusters dominating at successive time points, supporting the biological coherence of the clustering.

**4.2.4 Pseudotime.** We employed the Slingshot algorithm [32] to compute the pseudotime of cells. PCA coordinates and clustering labels were used as input to the analysis. Based on the correspondence between clusters and time

points, we set cluster "1" as the starting point and cluster "6" as the endpoint. Pseudotime progression was consistent with the actual sampling times, as shown by the violin plots of the pseudotime values for the cells in each time point separately (S5B Fig).

**4.2.5 Gene selection.** The dataset after pseudotime inference includes 14,841 genes and 758 cells. We focused our analysis on transcription factors relevant to the differentiation process. First, we selected genes from the dataset also found in a curated list of known human TFs by Vaquerizas et al. [50].

Differential expression analysis was then performed using the findMarkers function from the scran R package, applying the Wilcoxon rank-sum test across clusters. Markers were selected for each cluster by identifying transcription factors that showed statistically significant upregulation in at least one pairwise comparison against other clusters. Specifically, we retained genes that satisfied the thresholds of $p-value < 0.05$, $FDR < 0.05$, and $AUC > 0.92$ in at least one of the comparison of each cluster against all other cluster. This ensures that each selected gene is both statistically significant and shows strong discriminatory power in at least one comparison. The final set of TFs comprises 92 genes.

The genes SALL1, CXCR4, GLIS1, HNF4A, FOXA2, and SOX17 were manually added due to their known biological relevance in the context of endoderm differentiation, as supported by previous literature [31,51]. The final set of genes thus comprised 98 transcription factors.

**4.2.6 Mini-Bulk.** We applied the same Mini-Bulk (MB) strategy used for the mouse dataset to the human scRNA-seq data. Here we set the window size to 50 and the step size to one. The resulting dataset includes 98 transcriptional regulators and 709 cells (S5E Fig), and was used as input for the GRN inference with IGNITE.

## 4.3 IGNITE

We assumed that the activity of the genes can be effectively captured by a kinetic Ising model, governed by a simple spin-flip dynamics known as Glauber dynamics [19]. To infer the GRN interactions, we assessed the inverse problem for this model. Following the work of Zeng et al. [20], we further assume that the spins are updated asynchronously, and that the whole GRN is a fully connected network, with $N_{genes}^2$ possible interactions.

**4.3.1 Gene activity: Binarisation of the inputs.** We represent genes as a set of spins $\boldsymbol{s} = \{s_i\}_{i=1,\ldots,N_{genes}}$, where each $s_i$ is a binary variable that can assume values +1 and −1. Therefore, in order to infer the GRN from an Ising model and run IGNITE, it is necessary first to binarise transcriptomic data. We started from the input dataset (log-normalized scRNA-seq matrix with PST and MB). If we denote with $x_i(t)$ the gene expression of gene $i$ for the cell with PST value $t$, the binarisation of $x_i(t)$ was performed as follows:

$$s_i(t) = \begin{cases} +1 & \text{if} \quad x_i(t) \geq \frac{\max_t x_i(t)}{2} \\ -1 & \text{otherwise} \end{cases} \quad , \tag{2}$$

where $s_i(t) = +1$ if gene $i$ is active, and $s_i(t) = -1$ if it is inactive. Based on gene expression, this approach computes gene activity (GA) as a binary measure of activity, assigning a value of −1 for inactive genes and +1 for active genes. We tested this approach by verifying that naïve genes are mainly active at low PST values, whereas formative genes are mainly active during later steps in the pseudotime temporal evolution computing the heatmap of GA (S1H Fig).

**4.3.2 Asymmetric kinetic ising model.** The kinetic Ising model is specified by a coupling matrix of entries $J_{ij}$, which quantify gene-to-gene interactions and describes the GRN, and by a set of external fields $\boldsymbol{h} = \{h_i\}$, which measure the bias for a given gene to be active ($h_i > 0$) or inactive ($h_i < 0$). From the timeseries of $s_i(t)$, the goal of IGNITE is to infer $J_{ij}$ and $h_i$, which are assumed to be constant in time. Both $h_i$ and $J_{ij}$ are expressed in units of temperature.

Consider a time discretization with steps of size $\delta t$. During the time evolution of the model, the spins $s_i$ are updated asynchronously. At each time step $t$, a spin $i$ is randomly selected with probability $\gamma_i \delta t$, and its value $s_i$ is updated according to the transition probability:

$$p(s_i(t + \delta t)|s_i(t)) = \frac{\exp[s_i(t + \delta t)\theta_i(t)]}{2\cosh\theta_i(t)},$$

(3)

where $p(s_i(t + \delta t)|s_i(t))$ is the probability of the new spin value at time $t + \delta t$ given the one at the previous time $t$, and $\theta_i(t) = h_i + \sum_{j\neq i} J_{ij}s_j(t)$ is the effective local field acting on the $i$-th spin at time $t$.

### 4.3.3 Model reconstruction from data.

In principle, to perform the inference, we would need to know both the spin values at each time, $\mathbf{s} = \{s_i(t)\}$, and the update times, $\boldsymbol{\tau} = \{\tau_i\}$, for a given time interval $T$ with timestep $\delta t$. Then, starting from the spin history, our goal is to reconstruct the model parameters $h_i$ and $J_{ij}$. We can determine the learning rules for these parameters by likelihood maximisation [20]. We can write the likelihood as:

$$L(\{h_i\}, \{J_{ij}\}) = P(\mathbf{s}, \boldsymbol{\tau}) = P(\mathbf{s}|\boldsymbol{\tau})p(\boldsymbol{\tau}).$$

(4)

For each spin, the update times are a discretised Poisson process. This means that every time step $t$ is an element of the update time set $\boldsymbol{\tau}$ with probability $\gamma_i\delta t$. Therefore, the update time probability $p(\boldsymbol{\tau})$ is independent of the model parameters that we want to infer. For simplicity, we take $\gamma_i = \gamma$, assuming that $\gamma$ is known a priori, so that we do not need to determine it during the inference problem. For simplicity, we set $\gamma = 1$. Hence, the likelihood maximisation can be carried out over $P(\mathbf{s}|\boldsymbol{\tau})$ only.

Marginalising out the update times, we can obtain a simpler form of the update probability for the $i$-th spin, which is exactly the Glauber dynamics update rule:

$$p(s_i(t + dt)|s_i(t)) = \frac{\gamma\delta t}{2} \cdot (1 - s_i(t)\tanh(\theta_i(t)).$$

(5)

Then, the logarithm of the likelihood, i.e., the log-likelihood, is:

$$\mathcal{L}(\{h_i\}, \{J_{ij}\}) = \sum_i^{N_{\text{genes}}} \sum_{\tau_i}^{N_{\tau_i}} \left[ s_i(\tau_i + \delta t)\theta_i(\tau_i) - \log 2\cosh\theta_i(\tau_i) \right],$$

(6)

where $N_{\tau_i}$ is the number of update times for the gene $i$. We stress that maximising the log-likelihood is equivalent to maximising the likelihood. By taking a partial derivative of Eq 6, we obtain the following learning rule for the interaction matrix:

$$\delta J_{ij} \propto \frac{\partial\mathcal{L}(\{h_i\}, \{J_{ij}\})}{\partial J_{ij}} = \sum_{\tau_i}[(s_i(\tau_i + \delta t) - \tanh\theta_i(\tau_i))s_j(\tau_i)].$$

(7)

Importantly, if we define $J_{i0} = h_i$ and $s_0(t)=1$, Eq 7 includes also the learning rule for fields $h_i$ [20].

Crucially, Eq (7) requires the knowledge of the update times for the spins, which we typically cannot access. Indeed, in each time interval, one spin is randomly chosen for updating, but it does not necessarily flip. Thus, it is not generally possible to measure the update times from the timeseries only. To overcome this problem, we can proceed as in the work by Zeng et al. [20]. We define the correlation function $C_{ij}(t) \equiv \langle s_i(t_0 + t)s_j(t_0)\rangle$, where $\langle\cdot\rangle$ is the average over all realisations of the stochastic dynamics. The time derivative of the correlation function is:

$$\dot{C}_{ij}(t) = \lim_{dt\to 0} \frac{\langle s_i(t_0 + t + dt)s_j(t_0)\rangle_{t_0} - \langle s_i(t_0 + t)s_j(t_0)\rangle_{t_0}}{dt}.$$

(8)

By noting that we can separate the terms with and without spin flips, we can write:

$$\dot{C}_{ij}(t) = \frac{\langle s_i(t_0 + t + dt)s_j(t_0)\rangle_{t_0} - \langle s_i(t_0 + t)s_j(t_0)\rangle_{t_0}}{dt}$$

$$= \gamma dt \frac{\langle s_i(\tau_i + t + dt)s_j(\tau_i)\rangle_{\tau_i} - \langle s_i(\tau_i + t)s_j(\tau_i)\rangle_{\tau_i}}{dt} + (1 - \gamma dt)(\dots)_{\text{no flip}}$$

The term without flip vanishes because the spin did not update, and therefore $s_i(t + dt) = s_i(t)$. Hence, since $\delta t$ is the smallest time interval available from experimental trajectories, we can write:

$$\dot{C}_{ij}(0) = \gamma \left[ \langle s_i(\tau_i + \delta t)s_j(\tau_i)\rangle_{\tau_i} - C_{ij}(0) \right], \tag{9}$$

so that

$$\langle s_i(\tau_i + \delta t)s_j(\tau_i)\rangle_{\tau_i} = \frac{\dot{C}_{ij}(0)}{\gamma} + C_{ij}(0). \tag{10}$$

The updating rule (Eq 7) can be rewritten by substituting the first term of the right part of the equation with Eq 10:

$$\frac{\partial \mathcal{L}(\{h_i\}, \{J_{ij}\})}{\partial J_{ij}} = \frac{\dot{C}_{ij}(0)}{\gamma} + C_{ij}(0) - \langle \tanh \theta_i(t)s_j(t)\rangle, \tag{11}$$

where the second term in the right part of Eq 11 is an average over all time points. We note that this term is equivalent to the average over $\tau_i$ in Eq 7, since $\tanh \theta_i(t)s_j(t)$ is insensitive to whether an update has been made. Since the choice of the unit of time intervals is not relevant for inference, we set $\delta t = 1$ without loss of generality. Although convergence to the global maximum cannot be guaranteed, the update rule in Eq (11) will reach a local maximum that corresponds to a coupling matrix representing a functional GRN.

We note that the interaction matrix $J$ includes self-couplings for all genes. However, measuring mRNA does not allow us to distinguish between target and regulator genes. Thus, self-couplings lack direct biological relevance. However, this computational inclusion, while not reflecting direct biological significance, stabilises the inference process by introducing an informative bias that could compensate for specific dynamics or interactions not directly captured by the model. For this reason, we infer the self-couplings, and then remove them from the GRN without further investigation.

**4.3.4 Optimisation algorithms.** To find the maximum of Eq 11, we implemented two optimisation methods, the Momentum Gradient Ascent (MGA) algorithm (implemented as the maximisation version of the Stochastic Gradient Descent algorithm [57]) and the Nesterov-accelerated adaptive moment estimation algorithm (NADAM) [58]. In general, the maximisation of the likelihood is performed starting from an estimate of $J_{ij}$ and $h_i$, and updating this estimate with a small value in the direction of the gradient. This is repeated until convergence. Each loop of the learning rule is called an epoch.

**Momentum gradient ascent algorithm.** For the MGA algorithm, we adopted a step-decaying learning rate and an $L_1$ regularisation. The learning rate for the $n$-th epoch is:

$$\eta^n = \eta_0 \, dr^{\text{floor}(\frac{n}{n_{dr}})} \tag{12}$$

with $\eta_0$ initial learning rate, and $dr$ drop of the learning rate every $n_{dr}$ epochs. The $n$-th optimisation step for the fields can be written as:

$$v_{h,i}^{(n+1)} = \xi_m v_{h,i}^{(n)} + \eta_h^{(n)} \left[ \langle s_i(t) - \tanh \theta_i(t)\rangle - \lambda \text{sign}(h_i) \right] \tag{13}$$

$$h_i^{(n+1)} = h_i^{(n)} + v_{h,i}^{(n+1)}, \tag{14}$$

and the *n*-th optimisation step for the couplings is:

$$v_{J,ij}^{(n+1)} = \xi_m v_{J,ij}^{(n)} + \eta_J^{(n)} \left[ \frac{\dot{C}_{ij}(0)}{\gamma} + C_{ij}(0) - \langle s_j(t) \tanh \theta_i(t) \rangle - \lambda \mathrm{sign}(J_{ij}) \right] \tag{15}$$

$$J_{ij}^{(n+1)} = J_{ij}^{(n)} + v_{J,ij}^{(n+1)}. \tag{16}$$

The parameter $\xi_m$ in Eqs 13 and 15 is the momentum parameter, with $0 \leq \xi_m \leq 1$, and $\lambda$ is the regularisation parameter.

**Nesterov-accelerated adaptive moment estimation algorithm.** With the NADAM algorithm, we implemented both a step-decaying learning rate (Eq 12), and $L_2$ regularisation. To compute the first-moment $m$ and the second-moment $v$ of the gradient with respect to the coupling and the fields, we can write the *n*-th optimisation step as follows:

$$m_i^{(n+1)} = \beta_1 m_i^{(n)} + (1 - \beta_1) g_i^{(n+1)} \tag{17}$$

$$v_i^{(n+1)} = \beta_2 m_i^{(n)} + (1 - \beta_2)(g_i^{(n+1)})^2, \tag{18}$$

where $g_i$ is either $\frac{\partial \mathcal{L}(\{h_i\},\{J_{ij}\})}{\partial J_{ij}}$ or $\frac{\partial \mathcal{L}(\{h_i\},\{J_{ij}\})}{\partial h_i}$ (see Eq 11). To perform the learning step, we need to introduce the bias-corrected moments:

$$\hat{m}_i^{(n)} = \frac{m_i^{(n)}}{1 - \beta_1^n} \tag{19}$$

$$\hat{v}_i^{(n)} = \frac{v_i^{(n)}}{1 - \beta_2^n}, \tag{20}$$

so that:

$$h_i^{(n+1)} = h_i^{(n)} + \frac{\eta_h^{(n)}}{\sqrt{\hat{v}_h^{(n)}} + \epsilon} \left[ \beta_1 \hat{m}_h^{(n)} + \frac{(1 - \beta_1) g_{h_i}^{(n)}}{(1 - \beta_i^{(n)})} - \lambda \mathrm{sign}(h_i) \right] \tag{21}$$

$$J_{ij}^{(n+1)} = J_{ij}^{(n)} + \frac{\eta_J^{(n)}}{\sqrt{\hat{v}_J^{(n)}} + \epsilon} \left[ \beta_1 \hat{m}_J^{(n)} + \frac{(1 - \beta_1) g_{J_{ij}}^{(n)}}{(1 - \beta_i^{(n)})} - \lambda \mathrm{sign}(J_{ij}) \right], \tag{22}$$

where $\lambda$ is the *L2* regularisation parameter, and $\beta_1 = 0.9$, $\beta_2 = 0.999$ and $\epsilon = 10^{-8}$ are fixed parameters of the NADAM algorithm.

**4.3.5 Hyperparameters of the optimisation algorithms.** To perform the GRN reconstruction from Eq 11 we have to set the following hyperparameters in both the MGA and the NADAM optimisation algorithms.

- For the MGA optimizer: the step-decaying learning rate parameters, $\eta$, $dr$ and $n_{dr}$, the momentum parameter $\xi_m$, and the $L1$ penalty term $\lambda$.

- For the NADAM optimiser: the step-decaying learning rate parameters, $\eta$, $dr$ and $n_{dr}$, and the $L2$ penalty term $\lambda$.

We also note that the optimisation problem is not convex, as we expect Eq 6 to be characterised by many local maxima. To avoid sub-optimal solutions, we performed a random search among $N_{trial} = 250$ randomly selected sets of hyperparameters. The possible values considered for the hyperparameters are shown in the Table 7. The best set of hyperparameters can be defined with or without using prior knowledge.

1. With prior knowledge: we know a set of 18 interactions and we select the set of hyperparameters for which the measured FCI is maximum. If there are models that have equal FCI, we randomly select one of these models.

2. Without prior knowledge: we select the set of hyperparameters for which the measured CMD is minimum. If there are models that have equal CMD, we select randomly one of these models.

**4.3.6 Comparison with null models.** We conducted a permutation test to assess the statistical significance of the inferred interactions. The permutation test involved shuffling the row and column indices of the gene expression data $N_{test} = 50$ times and then evaluating $N_{sets} = 50$ sets of hyperparameters for each test dataset, to ensure thorough testing of different hyperparameter configurations. The final number of inferred null GRN with IGNITE is 2500. We investigated the statistical significance of the GRN inferred using IGNITE. We hypothesised that a network derived by chance would exhibit interaction values similar to those of null models. We assessed the 12 known interactions correctly inferred with IGNITE by comparing their values from the actual GRN against the probability distributions of these null model GRNs. Specifically, we checked if the values predicted by IGNITE for these interactions lay outside the 5th to 95th percentile range of the shuffled data. Since only three interactions aligned with the null model expectations (Nanog-Otx2, Stat3-Gbx2 and Tcf7l1-Esrrb), we reject the null hypothesis, supporting the statistical significance of the IGNITE GRN.

## 4.4 MaxEnt reconstruction

The element $\rho_{ij}$ between the genes $i$ and $j$ of the MaxEnt interaction matrix can be calculated as:

$$\rho_{ij} = -\frac{(C^{-1})_{ij}}{\sqrt{(C^{-1})_{ii}(C^{-1})_{jj}}},$$

(23)

where $C^{-1}$ is the inverse matrix of the covariance matrix calculated starting from our transcriptomics datasets, as described in [16,17]. The interaction matrix $\rho$ is symmetric.

**Table 7. Set of hyperparameters values for the random search. The momentum parameter is used only if the MGA optimiser is selected.**

| hyperparameters | values |
|---|---|
| optimiser | [MGA, NADAM] |
| $\eta_0$ | $\left[0.3, 0.4, 0.5, 0.6, 0.7, 0.8\right]$ |
| $dr$ | $\left[0.65, 0.70, 0.75, 0.80, 0.85, 0.90, 0.95, 0.99\right]$ |
| $n_{dr}$ | $\left[15, 20, 25, 30, 35\right]$ |
| $\xi_m$ (only for MGA) | $\left[0.75, 0.8, 0.85, 0.90, 0.95, 0.99\right]$ |
| $\lambda$ | $\left[0.01, 0.02, 0.03, 0.04, 0.05, 0.07, 0.09\right]$ |
| $N_{epochs}$ | [500,700,800,900,1200] |

## 4.5 SCODE reconstruction

SCODE [18] is a method for inferring regulatory networks from scRNA-Seq data designed for differentiating PSCs. It utilises linear ordinary differential equations (ODEs) and linear regression to capture the observed gene expression dynamics. We selected this method as the gold standard method comparable to IGNITE, because, as highlighted in [10], it has characteristics similar to those of IGNITE: (i) it needs time-ordered cells to capture the network dynamics, (ii) the inferred interactions are directed (asymmetric interaction matrix) and (iii) signed, and (iv) it is possible to generate new data in wild-type or under perturbation conditions. Looking at the other models evaluated in [10], SCODE is the model with these characteristics and with the best accuracy results for the experimental scRNA-seq datasets.

SCODE efficiency is based on the fundamental concept that the patterns of expression dynamics are finite and that expression dynamics can be accurately reconstructed using a limited number of these patterns. Therefore, the expression dynamic vector should be reduced from a length of $N_{genes}$ genes to a length $D$, with $D \ll N_{genes}$. This number $D$ is a parameter of the model and its value should be optimised, as suggested by the authors. To do that, we divided the dataset in training ($N_{genes} \times N_{cells}$ = 24 × 7638) and test datasets ($N_{genes} \times N_{cells}$ = 24 × 1909, 20% of the total dataset cells). We applied SCODE to training data and we evaluated the validity of the optimised model by computing the residual sum of squares (RSS) of the test dataset for various values of D ($D$ = 2, 4, 6 and 8). For each D, as in the original work, we executed SCODE 100 times independently. We observed that the median of the RSS values was almost saturated at $D$ = 6. As suggested by the authors, we checked the reproducibility of the inferred interaction matrix. The correlation coefficient was calculated among the optimised interaction matrices for the 50 replicates with the lowest RSS values, using the test data for each D. The correlations among replicates are high till $D$ = 6 and therefore an optimised interaction matrix remains stable till $D$ = 6. Considering the saturation of RSS values for the test data and the stability of the estimated interaction matrices at $D$ = 6, we selected this value for our analyses. The final inferred GRN interaction matrix is the element-by-element mean of the interaction matrices of the top 50 replicates.

## 4.6 CellOracle application

CellOracle [8] is a machine-learning framework that leverages GRNs inferred from single-cell multi-omics data to simulate gene perturbations and their effects on cell identity. The in silico perturbation unfolds in four steps: (1) Generation of cell-type or cell-state specific GRNs through cluster-wise regularised linear regression on multi-omics data. (2) Estimation of target gene expression changes due to transcription factor (TF) perturbations, using GRN models to simulate the cascading effects of these changes. CellOracle perturbation modelling is deterministic. (3) Calculation of cell-identity transition probabilities by comparing the gene expression shift with that of neighboring cells. (4) Transformation of these probabilities into a weighted vector, simplifying the complex gene expression shift into a two-dimensional representation to predict cell-state transitions post-TF perturbation. To compare CellOralce with IGNITE and SCODE, we focused on the first two steps of CellOracle, obtaining as output the inferred GRNs and the simulated gene expression after KO perturbations. CellOracle, as it is designed, can generate only perturbed gene expression data, not WT data. CellOracle requires as input the proceed scRNA-seq dataset with dimensionality reduction and clustering implemented. We took as input the scRNA-seq processed as described in 5.1 till the removal of the 2CLCs cluster. The processed dataset has 9696 cells and 2078 genes. These genes, identified as significant in the differentiation process by Carbognin et al. [14], were chosen over highly variable ones due to their established relevance to the process under study. We used as base GRN a fully connected network, inferring with CellOracle the weights of these connections (excluding self-loops, that are not inferred with CellOracle). We did not use the prebuilt GRN as suggested by the authors because in this network most of the interactions known from the literature are not present (78%). Then, we generated perturbed KO data using the function of the CellOracle Python package `simulate_shift`. After GRN inference and data generation, we focused only on 24 genes deemed critical for our study, along with their respective GRN.

## 4.7 Fraction of Correctly Inferred Interactions (FCI)

The Fraction of Correctly Inferred Interactions (FCI) is computed from the inferred interaction values. The FCI is given by:

$$FCI = \frac{\text{number of known interactions that were correctly inferred}}{\text{total number of known interactions}}.$$

(24)

The numerator is determined by counting the instances in which the sign of the inferred interaction values matches the interaction signs known from the literature. The denominator is equal to 18.

## 4.8 Generation and evaluation of WT data

**4.8.1 Gene activity generation with IGNITE.** We generated gene activity data using the interaction matrix inferred with the IGNITE method. The genes in this dataset, as for the IGNITE inputs, were represented as spins. To generate a new dataset, we leveraged the Ising model parameters, the inferred interaction matrix ($J$) and external fields ($h$), to determine the gene states for different cells by using the Glauber spins update rule. The resulting dataset is a matrix with the structure of the original data, with rows representing genes and columns representing distinct cells corresponding to different time steps. The data dimensions were equal to the input data ones ($N_{genes} \times N_{cells} = 24 \times 9547$).

To generate the data, we initialised the gene expression state vector for each gene in the network with random spin values. At each time step, we updated the values of each spin independently based on its interactions with other spins. Following Glauber dynamics, every spin can flip with probability $p = \frac{\gamma \delta t}{2} \cdot (1 - s_i(t) \tanh(\theta_i(t))$, with $\gamma = 1$, $\delta t = 1$, $s_i(t)$ current spin state of gene $i$ at time $t$, and $\theta_i(t) = h_i + \sum_{j \neq i} J_{ij} s_j(t)$ total field acting on spin $i$. Each newly generated dataset is distinct from the others, as IGNITE simulates gene activity via a stochastic dynamics.

**4.8.2 Gene expression generation with SCODE.** We employed the code provided by the authors of SCODE to reconstruct the expression. We generated a new dataset, capturing the dynamics of the 24 selected genes over 100 time steps, as suggested by the authors. The initial gene expression values for each gene were set to the mean expression of the first 1000 cells from the original log-normalized scRNA-seq dataset with PST and MB. We generated only one dataset since the data generation process is deterministic, ensuring that all generated datasets would be identical.

**4.8.3 Correlation matrices distance.** We assessed the similarity between the generated WT datasets (using IGNITE or SCODE) and the input dataset as follows.

i. Using IGNITE we generated $N_{trial} = 250$ gene activity datasets with dimensions identical to the original gene activity dataset ($N_{genes} = 24$ and $N_{cells} = 9547$). SCODE is deterministic, therefore we generated only one gene expression dataset with the number of cells suggested by the authors ($N_{genes} = 24$ and $N_{cells} = 100$).

ii. We computed the Pearson correlation matrices for the input datasets and for each of the simulated ones.

iii. For IGNITE only, we computed the element-by-element average of the $N_{trial}$ correlation matrices from the simulated data, obtaining an averaged correlation matrix.

iv. We computed the distance between the input data and the generated data matrices as follows:

$$d = \frac{1}{N_{genes}} \sqrt{\sum_{i}^{N_{genes}} \sum_{j}^{N_{genes}} (x_{ij} - x_{ij}^{sim})^2},$$

(25)

where $x_{ij}$ represents an element in the correlation matrix of the input gene activity/expression data, and $x_{ij}^{sim}$ represents an element in the correlation matrix (averaged for IGNITE) of the simulated data.

v. To scale distance measures $d$, we created $N_{shuffle} = 250$ random datasets by shuffling the input gene activity and gene expression datasets, as previously described for the IGNITE null model. We then scaled $d$ by dividing it by the average distance between the correlation matrix of the input data and the average correlation matrix of the null model datasets. We call the scaled $d$ quantity Correlation Matrices Distance (CMD).

A good inference method will produce simulated data with a CMD smaller than one. A value of CMD = 1 would imply that the simulated data correlation matrices are comparable to what could occur by chance. To quantify the goodness of IGNITE and SCODE results, we performed two statistical tests: (i) a two-sample t-test for IGNITE to compare the distribution of the $d$ values of the IGNITE generated data with those of the null model. (ii) the z-score to compare the single $d$ values obtained from SCODE with the distribution of the $d$ values for the null model.

## 4.9 Clustering of the data

We clustered the cells within a dataset by grouping cells based on similarities in their patterns. This allowed us to explore the underlying structure of the gene expression or gene activity datasets. We started with linkage analysis of the datasets using the scipy.cluster.hierarchy.linkage function from the SciPy library. We employed the linkage analysis with the Ward variance minimization algorithm. This is a hierarchical agglomerative approach aimed at minimizing the variance within each cluster. Subsequently, the results of the linkage analysis were visualised using a dendrogram and a heatmap. The heatmap provided an intuitive representation of gene expression/activity patterns across the cellular population. The dendrogram represents the hierarchical structure of the clusters, illustrating how individual cells or groups of cells are merged into larger clusters based on their gene expression/activity similarities (Fig 2D for input gene activity for IGNITE, Fig 2E for IGNITE generated gene activity, S2C Fig for input gene expressio for SCODE, and S2D Fig for SCODE generated gene expression).

## 4.10 Mouse perturbation of the system: gene knockout analysis

We perturbed the system by implementing single gene (for Rbpj, Etv5, or Tcf7l1) and triple gene (for Rbpj, Etv5, and Tcf7l1 simultaneously) knockout simulations to evaluate the prediction capability of IGNITE, SCODE, and CellOracle algorithms.

**4.10.1 Generation of KO data. With IGNITE:** To generate perturbed data after KOs of single or triple genes, we removed the rows and columns of the KO genes from the inferred interaction matrix. Then we generated $N_{trial} = 250$ gene activity datasets as described above (Method section 4.8.1). The datasets have $N_{genes} = 24 - N_{KO}$ genes, with $N_{KO}$ number of KO genes. The number of cells ($N_{cells} = 9547$) is identical to that of the input dataset.

**With SCODE:** We generated the simulated KO by removing the KO gene from the interaction matrix inferred with SCODE. Then we generated one gene expression dataset with $N_{genes} = 24 - N_{KO}$ for each KO, following the procedure described in the Method Section 4.8.2. The number of cells is $N_{cells} = 100$, as suggested by SCODE authors. We generated only one dataset since SCODE has deterministic dynamics.

**With CellOracle:** We generated the simulated KO data by using the specific function of the CellOracle Python package `simulate_shift`. As for SCODE, for each CellOracle KO, we generated a gene expression dataset with $N_{genes} = 24 - N_{KO}$. The number of cells is equal to the input dataset one ($N_{cells} = 9696$). We generated only one dataset since the perturbation procedure for CellOracle is deterministic.

**4.10.2 Measuring the influence of the KO gene on the GRN.** To assess the impact of gene knockout on gene activity or expression patterns, we compared the WT and KO datasets, considering KOs of single and triple genes. As WT data for IGNITE we generated 250 datasets as detailed in 4.8.1, and for SCODE we generated one WT dataset as described in 4.8.2. For CellOracle the WT dataset is the one used as input to infer the GRN, which is the original processed data with KNN-imputed values for dropout data. As input and generated data we have gene activity

when considering IGNITE data, and gene expression when considering SCODE and CellOracle data. Gene activity is comparable to the input data used for IGNITE: log-transformed scRNA-seq data with pseudotime, Mini-Bulk, and binarization applied. Conversely, gene expression data generated with SCODE and CellOracle are comparable to the input data required by these methods: log-transformed scRNA-seq with pseudotime and Mini-Bulk applied for SCODE, and log-transformed scRNA-seq for CellOracle.

To compare the WT datasets to the datasets of the KOs we removed the data of the perturbed genes. The quantification of this influence was carried out as follows.

**Simulated KO and WT data.** We first averaged the gene activity/expression levels per gene across all cells (and in all IGNITE-generated datasets). Next, we calculated the difference between KO and WT gene activity/expressions for each gene $i$, $\Delta x_i$:

$$\Delta x_i = x_i^{KO} - x_i^{WT}, \tag{26}$$

where $x_i^{KO}$ represents the average gene activity/expression of gene $i$ in the KO dataset, and $x_i^{WT}$ represents the same quantity in the WT dataset. A negative value of $\Delta x_i$ indicates that the KO perturbation causes a decrease in the average gene activity/expression of gene $i$, while a positive value suggests that KO causes an increase in the gene activity/expression for this gene. $\Delta x_i = 0$, instead, suggests that KO did not affect gene activity/expression of gene $i$.

**Experimental data.** Given the perturbation of a single gene $j$, or three genes $\{j,k,l\}$, to measure the effects of the perturbation on experimental data we employed the log2FC measures from [39]. A negative value signifies a reduction in gene $i$ expression due to the perturbation, while a positive value indicates an increase. Log2FC values near zero imply a negligible effect on gene $i$.

For the triple KO experimental data, we used the dataset from [38]. To quantify the perturbation impact on gene expression, we calculated the difference between two log2FC values for each gene: one comparing triple KO to control (2i) conditions, and another comparing gene expression in WT at 72h after the removal of 2i versus 0h. This is still a log2FC measure and improves the effects of the impact of KO on gene expression relative to the baseline expression in the WT condition.

### 4.10.3 Compare KO measures.
To ensure a meaningful comparison between the $\Delta x$ values of the simulated data and the log2FC values of the experimental data, we normalized both these quantities. For the generated data, all the $\Delta x$ values obtained from the four KO simulations were divided by the maximum absolute value of $\Delta x$, which was calculated across all $\Delta x$ values from all knockouts. Similarly, both single KO and triple KO Log2FC values were divided by their respective maximum absolute values. This normalization procedure ensures that all values are comparable and fall within the [–1, 1] interval. Notably, we avoided dividing the $\Delta x$ and log2FC values by their common absolute maximum value, as the distinct procedures used to generate the two datasets could make direct comparisons inappropriate. Moving forward, we will refer to both normalized values ($\Delta x$ and log2FC) as scaled KO-WT difference, $\widetilde{\Delta x}$, for clarity.

**Fraction of Agreement (FoA) for KO experiments.** To evaluate the concordance between the simulated and experimental KO-WT variation measures, $\widetilde{\Delta x}$, we defined the Fraction of Agreement (FoA). To compute the FoA, we set a threshold value thr = 0.05. Subsequently, we categorised the difference values into three classes: positive if $\widetilde{\Delta x_i} \leq$ thr = 0.05, negative if $\widetilde{\Delta x_i} \geq$ –thr, and null if $\widetilde{\Delta x_i}$ fall within the range (–thr, thr). Then we counted the instances in which the generated and experimental values of $\widetilde{\Delta x_i}$ for each gene $i$ matched in the categorical range (positive, negative, or null). This count was then divided by the total number of genes to obtain the FoA value ($N_{totalgenes} = N_{genes} - N_{KOgenes}$, equal to 23 for single KO simulations and to 21 for the triple KO).

To evaluate the statistical significance of the observed FoA values against random chance, we used a binomial distribution as a null model. The number of trials is $N_{trials} = N_{totalgenes}$. As before, we defined as success the instance where the generated and experimental $\widetilde{\Delta x_i}$ values for a gene fall within the same categorical range, with the potential number

of successes varying from 0 to $N_{trials}$. The probability of success in any given trial is $p = 1/3$. To determine the significance threshold for the FoA values, we calculated the minimum FoA value needed such that the cumulative probability of observing an FoA value that equal or higher, based on the binomial distribution, is less than or equal to 0.05. This threshold defines the value below which a FoA value is considered statistically significant against random chance. The determined threshold is $FoA_{thr} = 0.48$, for single and triple KO cases.

**Statistical analysis of KO prediction performance.** For each KO condition, Spearman correlations between simulated and experimental KO-WT differences were computed across the set of GRNs under evaluation (either the 10 GRNs with lowest CMD, the 10 GRNs with highest FCI, or the best GRN from 25 subsampled batches). Statistical significance was assessed by one-sided one-sample t-tests of the correlations against zero. P-values were adjusted for multiple testing across the four KO conditions using the Benjamini–Hochberg FDR procedure.

### 4.11 Human Perturbation of the system: gene knockout analysis

**4.11.1 Cluster areas estimation in PCA space.** To compute the areas corresponding to each cluster in PCA space (Fig 7C and 7D, and Fig 8A), we estimated smoothed cluster boundaries by computing local edge connections from Delaunay triangulation and retaining only those shorter than a fixed alpha threshold (we set $\alpha = 10$) [59]. The resulting set of edges was used to define an outer boundary, approximated via the convex hull of the retained segments.

To account for uncertainties in boundary positions—due to local sparsity or projection noise—we expanded each alpha shape by applying a geometric buffer of fixed width (0.5 units in PCA space). This buffer, computed as the Minkowski sum of the original polygon and a disk of radius equal to the buffer size, geometrically thickens the boundary, ensuring that edge cells are included within the cluster region. The buffer width was chosen empirically to balance two factors:(i) inclusion of meaningful boundary cells, and (ii) minimization of overlap between neighboring clusters. In cases of overlapping buffered regions, a predefined priority order was used to resolve conflicts. Cells not assigned to any region were labeled as "excluded".

**4.11.2 Fraction of cells per cluster for generated GA data.** We generated 150 dataset of WT gene activity with the same dimensions of the input WT gene activity data. Each dataset was processed independently, calculating the fraction of cells assigned to each cluster. Final cluster compositions were reported as the mean and standard error of the mean (SEM) across all generated datasets (Fig 7C and 7D).

**4.11.3 Comparison with KOs behavior from Chu et al.** To assess the impact of gene knockouts (KOs) on differentiation dynamics, we adopted a metric originally introduced by Chu et al., referred to as the Differentiation Score (DS). This score uses the the fraction of $CXCR4^+$ and $T^+$ cells in KO and WT conditions, two days after the onset of differentiation (approximately 45–48 hours). It is defined as:

$$DS = \frac{\left(\frac{f_{CXCR4}^{KO}}{f_T^{KO}}\right)}{\left(\frac{f_{CXCR4}^{WT}}{f_T^{WT}}\right)} \tag{27}$$

where $f_{CXCR4}^{KO}$ and $f_T^{KO}$ represent the fractions of $CXCR4^+$ and $T^+$ cells at 45–48 hours under the KO condition, and $f_{CXCR4}^{WT}$ and $f_T^{WT}$ are the corresponding fractions in the WT condition. A DS > 1 indicates an accelerated differentiation upon KO, whereas DS < 1 suggests a delayed or impaired differentiation.

We applied this metric to our IGNITE-generated KO data. To do so, we first identified the cluster in our dataset that most closely corresponds to day 2 of differentiation in Chu et al. work. We compared the proportion of cells expressing $T$ or $CXCR4$ (gene activity equal to 1) across clusters (S7A Fig). In the original study, 32.9% of WT cells were $CXCR4^+$ and 6.44% were $T^+$. In our data, cluster 4 best matched this profile. We therefore computed the Differentiation Score using WT and KO cells assigned to cluster 4.

#### 4.11.4 Comparison with POU5F1 KO behavior from Wang et al.
To validate our KO simulation results, we reanalysed microarray gene expression data from Wang et al. [51], available under GEO accession GSE34921. This dataset includes expression profiles for human PSCs under knock-out (shRNA-mediated) of pluripotency factors. Among the available KO conditions, we focused our comparison on the *POU5F1* KO, as this gene was both present in our inferred GRN and associated with a clear differentiation phenotype in the experimental dataset.

We focused on a curated list of genes involved in pluripotency and differentiation, corresponding to those shown in S6C Fig. Eight replicates, from day 1 to day 8 of differentiation, for both the WT and the *POU5F1* KO condition were used in the comparison. We computed the mean expression value across replicates for each gene and used it to calculate z-scores in S7B Fig.

## Supporting information

**S1 Fig. Preservation of gene expression dynamics following pseudotime and Mini-bulk processing.**
(PDF)

**S2 Fig. SCODE parameter optimization and validation of SCODE and CellOracle generated data against input mouse scRNA-seq.**
(PDF)

**S3 Fig. GRNs inferred with alternative methods on the mouse dataset.**
(PDF)

**S4 Fig. Statistical validation of inferred interactions against a null model.**
(PDF)

**S5 Fig. Gene expression patterns in human PSC differentiation.**
(PDF)

**S6 Fig. Inference and selection of the best-performing human GRN.**
(PDF)

**S7 Fig. Validation of IGNITE predictions against independent human knockout datasets.**
(PDF)

**S1 Table. Summary of benchmarking tasks for mPSC and hPSC datasets.**
(PDF)

## Author contributions

**Conceptualization:** Samir Suweis, Sandro Azaele, Graziano Martello.

**Data curation:** Merrit Romeike.

**Formal analysis:** Clelia Corridori.

**Funding acquisition:** Christa Buecker, Graziano Martello.

**Investigation:** Clelia Corridori, Merrit Romeike, Samir Suweis, Sandro Azaele, Graziano Martello.

**Methodology:** Clelia Corridori, Giorgio Nicoletti.

**Project administration:** Samir Suweis, Sandro Azaele, Graziano Martello.

**Resources:** Merrit Romeike, Christa Buecker.

**Software:** Clelia Corridori, Giorgio Nicoletti.

**Supervision:** Christa Buecker, Samir Suweis, Sandro Azaele, Graziano Martello.

**Visualization:** Clelia Corridori.

**Writing – original draft:** Clelia Corridori.

**Writing – review & editing:** Clelia Corridori, Giorgio Nicoletti, Christa Buecker, Samir Suweis, Sandro Azaele, Graziano Martello.

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
