## [Decision Letter · Decision Letter 0]

11 Jan 2026

PCOMPBIOL-D-25-02445

Unveiling Gene Perturbation Effects through Gene Regulatory Networks Inference from single-cell transcriptomic data

PLOS Computational Biology

Dear Dr. Martello,

Thank you for submitting your manuscript to PLOS Computational Biology. After careful consideration, we feel that it has merit but does not fully meet PLOS Computational Biology's publication criteria as it currently stands. Therefore, we invite you to submit a revised version of the manuscript that addresses the points raised during the review process.

We look forward to receiving your revised manuscript.

Kind regards,

Paolo Milazzo

Academic Editor

PLOS Computational Biology

Mark Alber

Section Editor

PLOS Computational Biology

**Additional Editor Comments:**

One of the reviewers asks for some improvements, mostly as regards presentation aspects and justification for some methodological choices. The authors should address these requests in a new version of the paper. The reviewer also suggests the authors to adopt a different packaging for the source code. This is a good suggestion to improve the usability of the code, but not mandatory for the acceptance of the paper.

**Journal Requirements:**

- TM on page: 15.

5) We have noticed that you have uploaded Supporting Information files, but you have not included a list of legends. Please add a full list of legends for your Supporting Information files after the references list.

1) State the initials, alongside each funding source, of each author to receive each grant. For example: "This work was supported by the National Institutes of Health (####### to AM; ###### to CJ) and the National Science Foundation (###### to AM).".

**Reviewers' comments:**

Reviewer's Responses to Questions

**Comments to the Authors:**

Reviewer #1: The authors have adequately addressed my concerns in this revised version of the manuscript.

Reviewer #2: My comments have been uploaded as an attachment.

**Have the authors made all data and (if applicable) computational code underlying the findings in their manuscript fully available?**

Reviewer #1: Yes

Reviewer #2: Yes

PLOS authors have the option to publish the peer review history of their article (what does this mean? ). If published, this will include your full peer review and any attached files.

**Do you want your identity to be public for this peer review?** For information about this choice, including consent withdrawal, please see our Privacy Policy .

Reviewer #1: No

Reviewer #2: No

**Figure resubmission:**
---

## [Editor Report · Decision Letter 1]

26 Feb 2026

Dear Professor Martello,

We are pleased to inform you that your manuscript 'Unveiling Gene Perturbation Effects through Gene Regulatory Networks Inference from single-cell transcriptomic data' has been provisionally accepted for publication in PLOS Computational Biology.

Best regards,

Paolo Milazzo

Academic Editor

PLOS Computational Biology

Mark Alber

Section Editor

PLOS Computational Biology

All reviewer comments have been fully addressed.